# Interferon-γ couples CD8⁺ T cell avidity and differentiation during infection

Lion F. K. Uhl [1], Han Cai[1], Sophia L. Oram[1], Jagdish N. Mahale[1], Andrew J. MacLean[1], Julie M. Mazet[1], Theo Piccirilli[1], Alexander J. He[1], Doreen Lau[2], Tim Elliott [2] & Audrey Gerard [1] ✉

Effective responses to intracellular pathogens are characterized by T cell clones with a broad affinity range for their cognate peptide and diverse functional phenotypes. How T cell clones are selected throughout the response to retain a breadth of avidities remains unclear. Here, we demonstrate that direct sensing of the cytokine IFN-γ by CD8⁺ T cells coordinates avidity and differentiation during infection. IFN-γ promotes the expansion of low-avidity T cells, allowing them to overcome the selective advantage of high-avidity T cells, whilst reinforcing high-avidity T cell entry into the memory pool, thus reducing the average avidity of the primary response and increasing that of the memory response. IFN-γ in this context is mainly provided by virtual memory T cells, an antigen-inexperienced subset with memory features. Overall, we propose that IFN-γ and virtual memory T cells fulfil a critical immunoregulatory role by enabling the coordination of T cell avidity and fate.

CD8⁺ T-cells are critical for clearing intracellular pathogens and tumors. The efficacy of CD8⁺ T-cell responses relies on the recruitment of different clones, the formation of a strong effector cytotoxic response, as well as the generation of memory cells that will provide enhanced protection upon rechallenge[1,2]. T-cell differentiation into effector or memory is regulated by multiple factors, including T-cell receptor (TCR) affinity for its cognate peptide, costimulatory signals and cytokines[3]. Despite considerable work, how those different factors are integrated to generate a robust albeit diverse short- and long-term response remains mostly unknown.

Avidity represents the capacity of T-cells to recognize infected cells and in turn elicit effector functions[4]. Peptide-MHC (p-MHC) tetramers, which bind multivalently and often predict functional responses, are often used as an approximate of the avidity of the endogenous T-cell response[5,6]. Quantification of the average avidity of the T-cell response is typically assessed by quantifying the up-regulation of activation markers (such as CD69) or effector functions (such as IFN-γ production) following stimulation of T-cells with increasing concentrations of cognate peptide[7,8]. Small variations of those avidity measures in the T-cell response towards a given pathogen can have drastic consequences. During COVID-19 infection,

hospitalized patients have only 3-4-fold lower average T-cell avidity to spike antigens than patients with milder disease[9]. Small differences in TCR avidity can also impact tolerance, as an increase in TCR avidity of <2-fold towards a self-antigen is enough to break tolerance in a mouse model of diabetes[10]. This indicates that the average avidity of an endogenous response has to be tightly regulated.

Together with co-receptors and co-stimulatory molecules, TCR affinity is an important factor controlling the avidity of the T-cell response[11]. High-affinity T-cell clones are sufficient to control infections[12]. They have a competitive advantage in interacting with antigen-presenting cells and expand better[13,14]. Despite these selective advantages, the presence of low-affinity T-cells are consistently observed[15–17] during an immune response, lowering the average avidity of the endogenous response, suggesting the existence of active regulatory mechanisms.

TCR affinity regulates CD8⁺ T-cell recruitment to an immune response, but also their fate. When a single CD8⁺ T-cell clone recognizes its cognate antigen with high-affinity, its fate is biased towards effector differentiation, whereas priming of the same clone with low-affinity peptide results in memory skewing[3,18]. However, a breadth of affinities, defined as the difference or extent between the lowest to the

[1]The Kennedy Institute of Rheumatology, University of Oxford, Oxford, UK. [2]Centre for Immuno-oncology, Nuffield Department of Medicine, University of Oxford, Oxford, UK. ✉e-mail: audrey.gerard@kennedy.ox.ac.uk

highest affinity clones, is found throughout immune responses[15,16], suggesting that clonal selection over differentiation states is actively regulated. Cells bearing the same TCR can lead to heterogenous differentiation patterns[1,19,20], further supporting the notion that additional factors regulate the relationship between differentiation and TCR affinity.

This overall suggests that regulation of T-cell avidity throughout an immune response requires the co-regulation of differentiation and avidity. Co-regulation implies that T-cells have to directly receive and integrate the multiple cues, as a system solely relying on tailoring the strength of extrinsic regulatory signals to the size of the population regulated would not provide robustness at a population level[21]. Some of the cues T-cells require to regulate their response are directly shared between T-cells[21–26]. Recently, a bioinformatic approach designed to identify mediators of T-cell communication highlighted IFN-γ as one of the top candidates[21]. IFN-γ controls CD8+ T-cell differentiation, expansion[22,23,27–29] and the immunodominance of the T-cell response[27]. Because IFN-γ regulates T-cell responses in part through direct signaling in T-cells[22,23,28,30], it is, therefore, a strong candidate for regulating avidity throughout an immune response.

Here, we present evidence that direct IFN-γ sensing by CD8+ T-cells regulates clonal selection, by coordinating avidity and differentiation during an immune response. By deleting the receptor of IFN-γ in CD8+ T-cells, we demonstrated that IFN-γ limits the expansion of T-cells exhibiting high avidity towards their cognate peptide, while increasing the expansion of T-cells with lower avidity and skewing their differentiation towards effector. As a result, IFN-γ lowers the avidity of the primary response, while increasing the avidity of the memory response. Importantly, the regulation of avidity by IFN-γ has profound consequences, resulting in sub-optimum primary immunity towards viral infection, a trade-off leading to improved memory responses. IFN-γ signaling in CD8+ T-cells occurs during priming, where it is provided by a specific CD8+ T-cell subset, called virtual memory T-cells (T_VM). We propose that T_VM cells fulfil a critical immunoregulatory role in modulating collective T-cell behavior. Overall, our data demonstrate that IFN-γ coordinates clonal selection during differentiation to allow for a retention of T-cell avidities throughout the immune response and a balance between short- and long-term immune responses.

## Results

### IFN-γ sensing by CD8+ T-cells results in sub-optimum immunity to Influenza

Given the tight regulation of T-cell responses in terms of avidity and differentiation, we hypothesized that inhibiting the factors intrinsically regulating these two features would have major consequences on CD8+ T-cell-dependent immune responses. Because IFN-γ has been implicated in controlling the immunodominance and differentiation during infection and can act directly on T-cells, we focused on IFN-γ and first determined the consequence of blocking IFN-γ sensing by CD8+ T-cells for viral responses. To this aim, we deleted the IFN-γR in CD8+ T-cells by crossing the E8I-Cre model[31] to IFN-γR1flox/flox mice[32] (CD8-IFN-γRKO). IFN-γR deletion was specific for CD8+ T-cells, as assessed by IFN-γR1 staining (Fig. S1a,b). This was confirmed by crossing Cd8a-Cre to ROSA-dtTomato mice, where dtTomato expression was used as a read-out of Cre expression (Fig. S1c). CD8-IFN-γRKO or control mice were infected with the Influenza virus strain X31 expressing OVA (X31-OVA) and their weight was monitored over time as a general read-out of control of the infection. Overall, CD8-IFN-γRKO mice lost less weight than control mice (Fig. 1a) following infection with a sub-lethal dose of X31-OVA, suggesting CD8-IFN-γRKO mice controlled infection better than control mice. This was confirmed by quantifying virus load over time by quantitative PCR, showing that CD8-IFN-γRKO mice cleared the virus faster than control mice (Fig. 1b, S1d). In addition, control mice exhibited impaired survival when infected with a higher dose of X31-OVA, while CD8-IFN-γRKO mice

recovered (Fig. 1c, d). We concluded that IFN-γ-sensing by CD8+ T-cells results in sub-optimum effector responses against Influenza.

We then investigated the mechanism by which IFN-γ regulated CD8+ T-cell responses to Influenza infection. We tested whether increased cytotoxicity was induced in CD8-IFN-γRKO mice. Using in vivo cytotoxicity assay, we found that killing was efficient in both control and CD8-IFN-γRKO mice (Fig. 1e, S1e). In addition, ex vivo surface expression of lysosome-associated membrane protein-1 (LAMP-1), a marker of cytotoxic CD8+ T-cell degranulation, was comparable between control and CD8-IFN-γRKO CD8+ T-cells (Fig. 1f). Similarly, we did not observe any difference in Perforin (Fig. 1g) and Granzyme B (Fig. 1h) expression between CD8-IFN-γRKO and control mice. This demonstrates that the sub-optimum CD8+ T-cell immunity induced by IFN-γ sensing was not the result of impaired intrinsic cytotoxic functions. Because IFN-γ is known to regulate expansion and trafficking[33,34], we then investigated whether the increased control by CD8+ T-cells was due to increased cell number in the lung. The number and fraction of lung OVA-specific CD8+ T-cells, analyzed by Kb-N4 tetramer staining (Fig. S1f) over time, was equivalent between CD8-IFN-γRKO and control T-cells (Fig. 1i, j), indicating that IFN-γ-sensing by CD8+ T-cells did not regulate their overall expansion or recruitment to effector sites.

As effector function and recruitment were largely unaffected by IFN-γ sensing, we then investigated whether IFN-γ sensing by CD8+ T-cells affected the average avidity of the T-cell response by analyzing Kb-N4 tetramer MFI of OVA-specific tetramer+ CD8+ T-cells. Differences in tetramer staining may simply reflect differences in TCR expression or TCR down-regulation following stimulation. To ensure that differences in tetramer binding between control and CD8-IFNγ RKO mice would not be the result of differential TCR expression, we normalized the MFI of tetramer with the one of CD3 or CD8. This has been used previously in other studies, a measure of TCR affinity and/or avidity at the population level[5], which we refer to as relative avidity. CD8-IFN-γRKO T-cells exhibited increased relative avidity (Fig. 1k), indicating that IFN-γ sensing by CD8+ T-cells lowers the avidity of primary responses, leading to sub-optimum immunity to Influenza.

### IFN-γ sensing by CD8+ T-cells decreases the avidity of the effector T-cell response

To explore whether the regulation of avidity by IFN-γ was a general mechanism, we switched to another model of infection, *Listeria monocytogenes* (LM) expressing Ovalbumin (OVA), an intracellular pathogen for which CD8+ T-cell response is well characterized. This model also allows us to alter the affinity and avidity of CD8+ T-cell responses by using LM expressing the dominant OVA peptide recognized by CD8+ T-cells (N4) or altered OVA peptides of lower affinity.

We first tested whether the avidity of CD8+ T-cell primary responses was also affected by IFN-γ sensing following LM infection. CD8-IFN-γRKO or control mice were infected with LM expressing OVA (LM-OVA) and OVA-specific CD8+ T-cells were analyzed by Kb-N4 tetramer staining. In this system, expansion of endogenous OVA-specific T-cells was increased in CD8-IFN-γRKO mice (Fig. 2a *and* S2a), most likely because the spleen is both a priming and effector site during LM infection. In addition, OVA-specific T-cells exhibited a significant shift in tetramer binding towards higher MFI (Fig. 2b, c). As for Influenza infection, normalizing tetramer binding by CD3 or TCR expression to extract relative avidity measures indicated that CD8-IFN-γRKO CD8+ T-cells were of higher avidity compared to their control counterparts (Fig. S2b–d). This was not a consequence of increased LM-OVA load (Fig. S2e), increased IFN-γ production (Fig. S2f) or differential priming, as assessed by CD69 upregulation (Fig. S2g). Increased tetramer binding in CD8-IFN-γRKO CD8+ T-cells was not present before 5 days after infection (Fig. S2h), suggesting that IFN-γ did not inhibit priming of high-affinity T-cells.

Our data indicate that IFN-γ-sensing by CD8+ T-cells lowers the avidity of the primary response in two different infection models. But

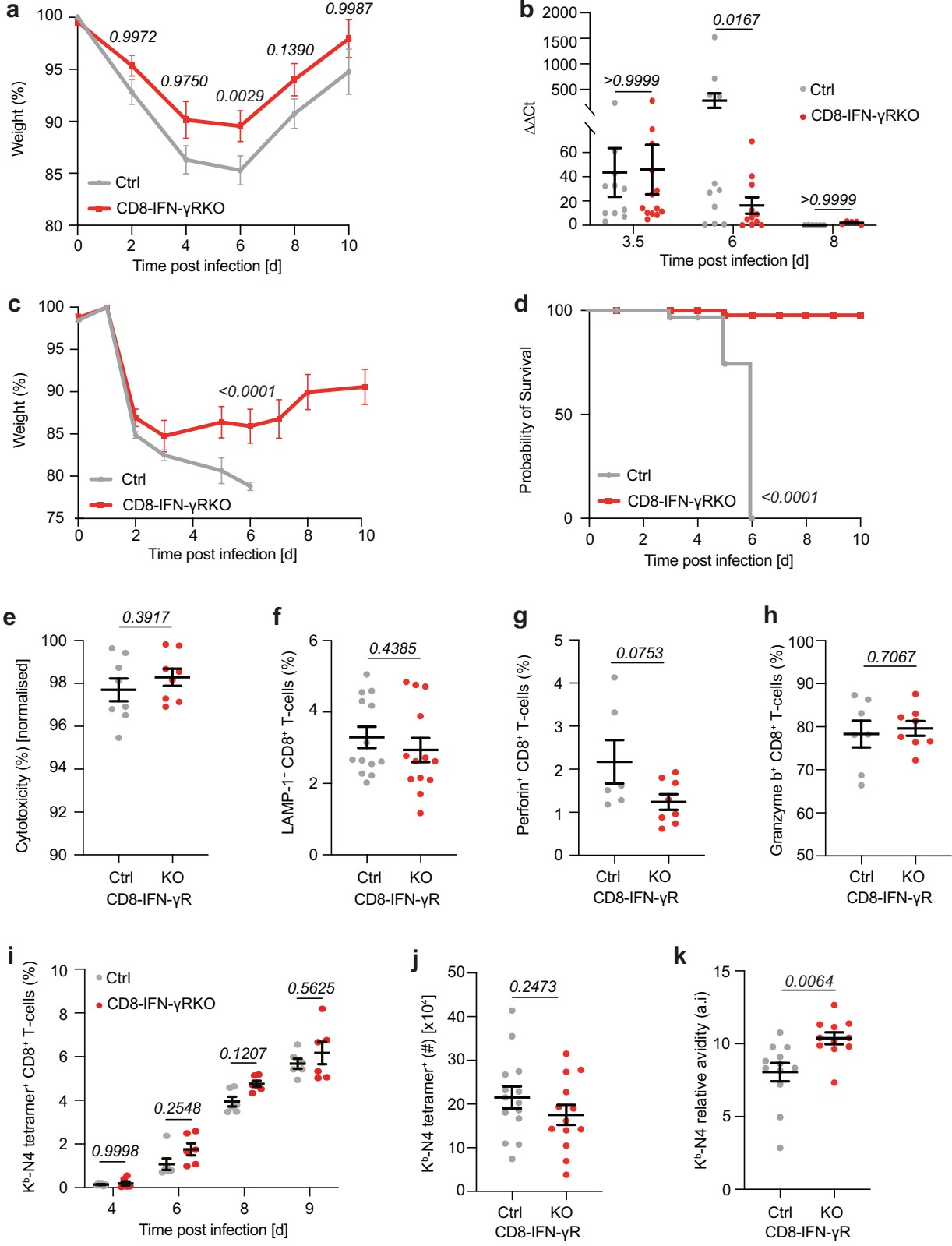

because the tetramer itself and the staining method can impact the range of discrimination between T-cell clones of different avidities[35], we then sought to quantify the difference in avidity between control and CD8-IFN-γR^KO CD8⁺ T-cell responses. We measured IFN-γ production following stimulation of CD8⁺ T-cells with increasing concentrations of the OVA peptide (N4) ex vivo. IFN-γ expression by control and CD8-IFN-γR^KO T-cells were similar in magnitude (Fig. 2d) but the EC₅₀

was decreased for CD8-IFN-γR^KO compared to control CD8⁺ T-cells, both by flow cytometry (Fig. 2e,f) and ELISA (Fig. 2g,h), indicating a higher avidity of the CD8-IFN-γR^KO T-cell repertoire. This demonstrated that IFN-γ-sensing by CD8⁺ T-cells decreases the avidity of the primary response by about 2–4-fold. We validated this result by quantifying OVA-specific CD8⁺ T-cell avidity by dynamic acoustic force measurements. Endogenous OVA-specific T-cells from control and CD8-IFN-

**Fig. 1 | IFN-γR deletion in CD8$^+$ T-cells improves effector responses against Influenza infection. a, b** CD8-IFN-γR$^{KO}$ (KO, red) and control (Ctrl, grey) mice were infected with $4 \times 10^4$ pfu X31-OVA. **a** Weight was measured every two days and quantified as relative weight loss ($n = 12$ Ctrl, 14 KO animals). **b** Lung viral titres were assessed by qRT-PCR on day 3/4 ($n = 11$ Ctrl, 12 KO animals), 6 ($n = 11$ animals) and 8 ($n = 6$ Ctrl, 5 KO animals). Expression levels were normalised to the housekeeping gene β-actin. **c, d** CD8-IFN-γR$^{KO}$ (KO, red) and control (Ctrl, grey) mice were infected with $4 \times 10^5$ pfu X31-OVA. Graphs show average weight ($n = 9$ Ctrl, 8 KO animals) (**c**) and survival ($n = 9$ animals) (**d**) over time. **e** CD8-IFN-γR$^{KO}$ (KO, red) and control (Ctrl, grey) mice were infected with LM-OVA and injected with ad-mixed N4-loaded or unloaded target splenocytes 7 day post-infection to quantify in vivo cytotoxicity. The graph shows in vivo cytotoxicity normalized by the percentage of endogenous OVA-specific T-cells ($n = 8$ animals). **f–h** CD8-IFN-γR$^{KO}$ (KO, red) and control (Ctrl, grey) mice were infected with LM-OVA and splenocytes were isolated after 7–9 days. **f** Splenocytes were re-stimulated with N4 peptide and LAMP-1 expression in CD8$^+$ T-cells was analyzed by flow cytometry and normalized by the percentage of endogenous OVA-specific T-cells ($n = 13$ animals). Perforin (**g**) and Granzyme B (**h**) expression was analyzed by flow cytometry ($n = 7$ Ctrl, 8 KO animals). **i–k** CD8-IFN-γR$^{KO}$ (KO, red) and control (Ctrl, grey) mice were infected with $4 \times 10^4$ pfu X31-OVA. **i** Relative abundance of lung N4-tetramer$^+$ CD8$^+$ T-cells was analysed by flow cytometry on day 4, 6, 8 ($n = 6$ animals) or 9 ($n = 14$ Ctrl, 13 KO animals). Absolute numbers ($n = 14$ Ctrl, 13 KO animals) (**j**) and relative avidity ($n = 12$ Ctrl, 11 KO animals) (**k**) of N4-tetramer$^+$ CD8$^+$ T-cells were analyzed by flow cytometry between day 7 and 10 post-infection. Data are from ≥3 (**a–f, i–k** or 2 **g, h**) independent experiments. Two-tailed unpaired Student's t-test (E-H, J-K), Two-way ANOVA with Šidák's multiple comparison test (**a–c**) Mantel-Cox test (**d**). Error bars indicate the mean ± s.e.m.

γR$^{KO}$ mice were isolated at the peak of LM-OVA infection and allowed to adhere on a flow chamber containing a monolayer of K$^{b+}$-OVA-expressing cells. Increasing acoustic force was then applied and T-cell detachment from the monolayer was monitored by microscopy. Using this method, we confirmed that CD8-IFN-γR$^{KO}$ OVA-specific T-cells displayed increased avidity compared with control T-cells (Fig. 2i,j). Taken together, we demonstrated that IFN-γ-sensing by CD8$^+$ T-cells decreases the avidity of the primary response using multiple techniques, showing that this phenotype is robust.

Altogether, we concluded that IFN-γ signaling in CD8$^+$ T-cells decreases the avidity of the effector response.

## CD8$^+$ T-cell paracrine IFN-γ signalling during priming regulates the avidity of the effector response

IFN-γ production during LM-OVA infection is characterized by an early wave of IFN-γ production occurring during priming and a second wave during the peak of the effector response[23,36,37] (Fig. S3a). To determine when IFN-γ was sensed by CD8$^+$ T-cells to regulate T-cell avidity, anti-IFN-γ was administered either 16–24 h or at day 5 and 6 post-infection. Similar to deleting IFN-γR on CD8$^+$ T-cells, blocking IFN-γ 16–24 h post-infection resulted in increased abundance of OVA-specific CD8$^+$ T-cells (Fig. 3a) and increased tetramer (but not CD3) staining (Fig. 3b,c). This data demonstrated that the increased avidity observed in CD8-IFN-γR$^{KO}$ mice was not the result of indirect increased IFN-γ signaling in other cells. Indeed, because the majority of cells in the spleen are T-cells, there was a possibility that IFN-γR deletion in T-cells resulted in increased IFN-γ bio-availability and thereby enhancing IFN-γ signaling in neighboring cells, indirectly increasing T-cell avidity. Taken together, we concluded that IFN-γ was sensed 16–24 h post-infection to regulate the expansion and avidity of OVA-specific CD8$^+$ T-cells. Blocking IFN-γ during the second wave had no effect on those measures.

Because NK cells were the main source of IFN-γ at the onset of LM infection (Fig.S3b), we investigated whether they contributed to the regulation of T-cell avidity. To do so, we ablated NK cells with depleting NK1.1 antibody (Fig. S3c) and infected mice with LM-OVA. The proportion and avidity of the T-cell response against OVA was assessed at the peak of the response. Depleting NK cells resulted in a slight increase in the proportion of tetramer$^+$ T-cells (Fig. 3d), most likely due to the increased bacterial load, as mice without NK cells fail to control LM infection[38]. However, this was not accompanied by an increase in tetramer staining (Fig. 3e), relative avidity (Fig. S3d) and functional avidity (Fig. 3f,g) demonstrating that NK cells were not the source of IFN-γ implicated in the regulation of T-cell avidity.

Because IFN-γ is shared between CD8$^+$ T-cells during priming[23] and the second cellular source of IFN-γ (Fig. S3b), we hypothesized that CD8$^+$ T-cell-derived IFN-γ may be the dominant source regulating CD8$^+$ T-cell avidity. To test this, we created mixed bone marrow (BM) chimeras by reconstituting lethally irradiated IFN-γ$^{KO}$ mice with ad-mixed IFN-γ$^{KO}$ and CD8α$^{KO}$ BM, resulting in IFN-γ deletion specifically in CD8$^+$ T-cells (CD8-IFN-γ$^{KO}$ mice). Reconstitution of IFN-γ$^{KO}$ mice with ad-mixed IFN-γ$^{KO}$ and WT BM was used to generate control (WT) mice (Fig. 3h). STAT1 phosphorylation in CD8$^+$ T-cells was analyzed 24 h post-LM-OVA infection, as a read-out of IFN-γ signaling. WT CD8$^+$ T-cells exhibited enhanced IFN-γ signaling compared to CD8-IFN-γ$^{KO}$ cells (Fig. 3i). Ablation of CD8-induced IFN-γ led to 40% inhibition in Stat1 phosphorylation (Fig. 3i) while CD8$^+$ T-cells constitute a small proportion of IFN-γ-producing cells (around 15%, Fig.S3b), suggesting that CD8$^+$ T-cells are highly sensitive to their own IFN-γ. At the peak of the response, we observed no increase in the proportion of OVA-specific T-cells (Fig. 3j) in CD8-IFN-γ$^{KO}$ mice, but enhanced tetramer binding (Fig. 3k), as observed with CD8-IFN-γR$^{KO}$ mice (Fig. 2c).

Taken together, we concluded that CD8$^+$ T-cell paracrine IFN-γ signaling regulates the avidity of their response.

## Virtual memory T (T$_{VM}$) cells constitute the CD8$^+$ T-cell subset producing IFN-γ during priming

Given that CD8$^+$ T-cells primarily sense their own IFN-γ to regulate their avidity, we then characterized which CD8$^+$ T-cell subset secreted IFN-γ using scRNA-seq of CD8$^+$ T-cells from control mice and mice infected with LM-OVA for 24 h. Unsupervised hierarchical clustering identified 6 clusters (Fig. S4a), which were labelled based on known markers of naïve and memory T-cells (Fig. S4b). We distinguished 4 clusters harboring a naïve phenotype that we merged (Figs. 4a,b), and 2 clusters with a memory phenotype (*Cd44, Eomes* and *Cxcr3*). One memory cluster had features of naïve cells, such as *Lef1, Ccr7* and *Sell* (encoding for CD62L) expression, expressed high levels of *Il2rb* and low levels of the integrin *Itga4* (Fig. 4a,b). These markers are characteristic of virtual memory T-cells (T$_{VM}$), a subset of antigen-independent memory T-cells[39]. The T$_{VM}$ characteristic expression pattern was confirmed by Differential Gene Expression analysis (Fig. 4c). Analysis of IFN-γ transcript revealed that T$_{VM}$ cells were the predominant subpopulation producing IFN-γ (Fig. 4d–f). We confirmed this by flow cytometry using GREAT mice, an IFN-γ reporter strain whereby YFP expression is driven by the IFN-γ promoter[40]. GREAT mice were infected with LM-OVA and the production of IFN-γ by the different CD8$^+$ T-cell subsets was assessed using CD44, CD122 and CD49d *(Itga4)* to delineate naïve, T$_{VM}$ and antigen-experienced memory CD8$^+$ T-cells. Up to 70% of IFN-γ producers were T$_{VM}$ (Fig. 4g) 24 h after LM-OVA infection, whereas they constitute less than 20% of CD8 T-cells (Fig. S4c). Similar data was observed with LM expressing gp33, ruling out the possibility of antigenic bias (Fig. 4g). We concluded that T$_{VM}$ was the main CD8$^+$ T-cell subset producing IFN-γ during priming.

We then wondered whether all CD8$^+$ T-cell subsets could sense IFN-γ. To answer this question, we used NicheNet[41] to analyze cell-cell communication between CD8$^+$ T-cell subsets. Interestingly, IFN-γ was the top predicted ligand received by all CD8$^+$ T-cells (Fig. S4d) or naïve CD8$^+$ T-cells (Fig. S4f). As already observed, T$_{VM}$ was predicted to be the main IFN-γ producer ("sender", Fig. S4e,g). This indicated that all CD8$^+$ T-cells could receive IFN-γ. This was confirmed by computing an IFN-γ signaling score, which shows that all CD8$^+$ T-cells sense IFN-γ,

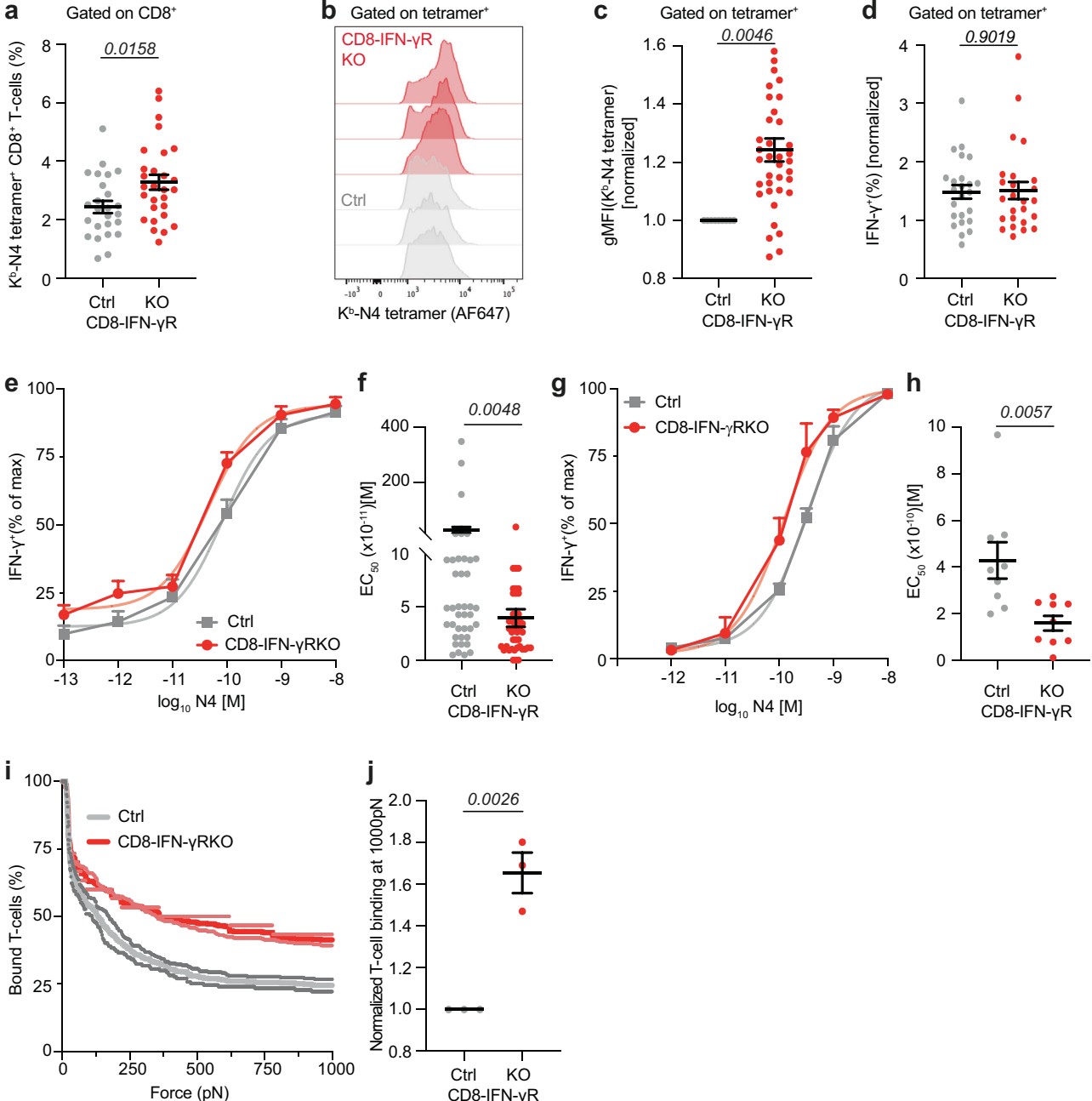

**Fig. 2 | IFN-γR deletion in CD8+ T-cells increases the avidity of effector T-cells.** **a**–**h** CD8-IFN-γR^KO (KO, red) and control (Ctrl, grey) mice were infected with LM-OVA and splenocytes were isolated after 7−10 days. Relative abundance (*n* = 25 Ctrl, 28 KO animals) (**a**), representative N4-tetramer histogram (**b**) and normalized N4-tetramer MFI (*n* = 37 animals) (**c**) of N4-tetramer+ CD8+ T-cells, analyzed by flow cytometry. N4-tetramer gMFIs of KO samples were normalized to the mean Ctrl N4-tetramer gMFI within each independent experiment. **d**–**h** Splenocytes were restimulated in vitro with the indicated concentrations of N4 for 16 h. **d** Percent KO and Ctrl CD8+ T-cells expressing IFN-γ at the maximum peptide concentration, assessed by flow cytometry and normalized by % of N4-tetramer+ CD8+ T cells (*n* = 24 Ctrl, 25 KO animals). Analysis of relative IFN-γ expression (**e**) and corresponding EC50 (**f**) by flow cytometry (*n* = 15 Ctrl, 16 KO animals). (**g, h**) Analysis of relative IFN-γ expression (**g**) and corresponding EC50 (**h**) by ELISA (n = 9 animals). **i, j** CD8-IFN-γR^KO (KO, red) and control (Ctrl, grey) mice were infected with LM-OVA and N4-tetramer+ CD8+ T-cells were sorted after 9 days to measure TCR avidity by acoustic force spectroscopy (*n* = 3 experiments, 4 mice per experiments). **i** Representative graph showing KO (red) and Ctrl (grey) CD8+ T-cell binding to target cells during acoustic force measurement. **j** Quantification of KO and Ctrl CD8+ T-cell binding at maximum force. Data are from 6 (**a**–**d**) or ≥ 3 (**e**–**j**) independent experiments. Two-tailed unpaired Student's *t*-test (**a**–**d**, **h**, **j**) and two-tailed Mann-Whitney test (**f**). Error bars indicate the mean ± s.e.m.

although T_VM to a lesser extent (Fig. S4h), most likely because they exhibit lower IFN-γR expression (Fig. S4i). Overall, this confirmed that T_VM are the main producers of IFN-γ, which is sensed by other CD8+ T-cells.

T_VM cells are known to exhibit enhanced sensitivity to inflammatory cytokines, resulting in bystander activation[42]. Differential pathway analysis, however, suggested that T_VM priming was more complex than bystander priming during LM-OVA infection. While true (antigen-experienced) memory T-cells displayed signatures of cytokine priming, T_VM exhibited additional signatures, some shared with naïve T-cells, including increased metabolism, heat-shock response, and cell cycle, which could indicate TCR priming (Fig. S5). To address whether IFN-γ production by T_VM required TCR priming, CD8+ T-cells from GREAT mice were activated in vitro with anti-CD3ε/-CD28 ("TCR

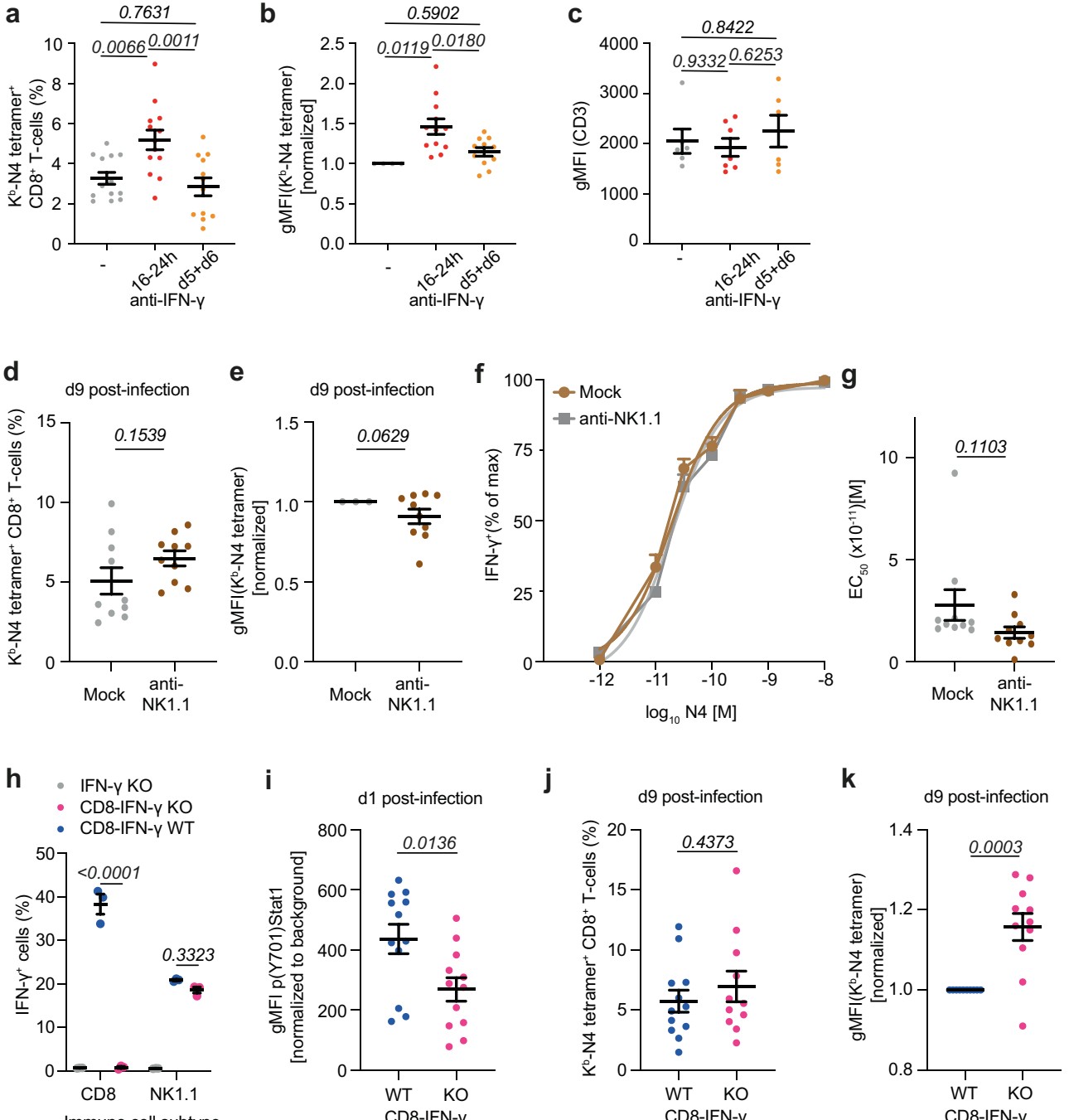

**Fig. 3 | IFN-γ·is by CD8+ T-cells at priming and is paracrine. a–c** WT mice were infected with LM-OVA and mice were either left untreated (Ctrl, grey) or treated with anti-IFN-γ 16-24 h (red) or at day 5 and 6 (orange) post-infection. Graphs show relative abundance (n = 13 Ctrl and 24h-anti-IFN-γ-treated animals, 12 day 5/6-anti-IFN-γ-treated animals) (**a**), normalized N4-tetramer gMFIs (n = 13 Ctrl/24h-anti-IFN-γ-treated animals, 12 day5/6-anti-IFN-γ-treated animals) (**b**), and CD3 gMFI (n = 6 Ctrl/24h-anti-IFN-γ-treated animals, 7 day5/6-anti-IFN-γ-treated animals) (**c**) of N4-tetramer+ CD8+ T-cells analyzed by flow cytometry 9 days post-infection. **d, e** WT mice were treated with depleting NK1.1 (brown) or control antibodies (grey) and infected with LM-OVA. **d, e** Relative abundance (**d**) and the normalized N4-tetramer gMFIs (**e**) of N4-tetramer+ CD8+ T-cells 9 days post-infection (n = 10 animals). **f–g** Splenocytes were re-stimulated in vitro with the indicated concentrations of N4 for 16 h. Quantification of relative IFN-γ expression (n = 5 animals) (**f**) and EC50 of IFN-γ production (n = 10 animals) (**g**) by flow cytometry. **h–k** Control (Ctrl, blue)

and CD8-IFN-γ[KO] (pink) chimera mice were infected with LM-OVA. **h, i** Splenocytes were isolated after 24 hrs. **h** IFN-γ production by CD8+ T-cells and NK cells was quantified after ex-vivo PMA and Ionomycin stimulation (n = 3 animals). **i** STAT1 phosphorylation (pSTAT1) was quantified in activated CD8+ T-cells (n = 12 animals). pSTAT1 of BM chimeras was normalized to IFN-γKO pSTAT1 MFI by subtracting the background gMFI. **j, k** Splenocytes were isolated after 9 days. Relative abundance (n = 11 Ctrl, 12 CD8-IFN-γ[KO] animals) (**j**) and normalized N4-tetramer gMFIs (n = 11 animals) (**k**) of N4-tetramer+ CD8+ T-cells. N4-tetramer gMFIs of IFN-γ- or NK-depleted samples were normalized to the mean Ctrl N4-tetramer gMFI within each independent experiment. Data are from ≥3 (**a–e, i–k**), 2 (**f, g**) or representative of 3 (**h**) independent experiments. Two-tailed unpaired Student's t-test (**d, e, g, j, k**), one-way ANOVA with Tukey's multiple comparison test (**a–c**) and two-way ANOVA and Šidák's multiple comparison test (**h**). Error bars indicate the mean ± s.e.m.

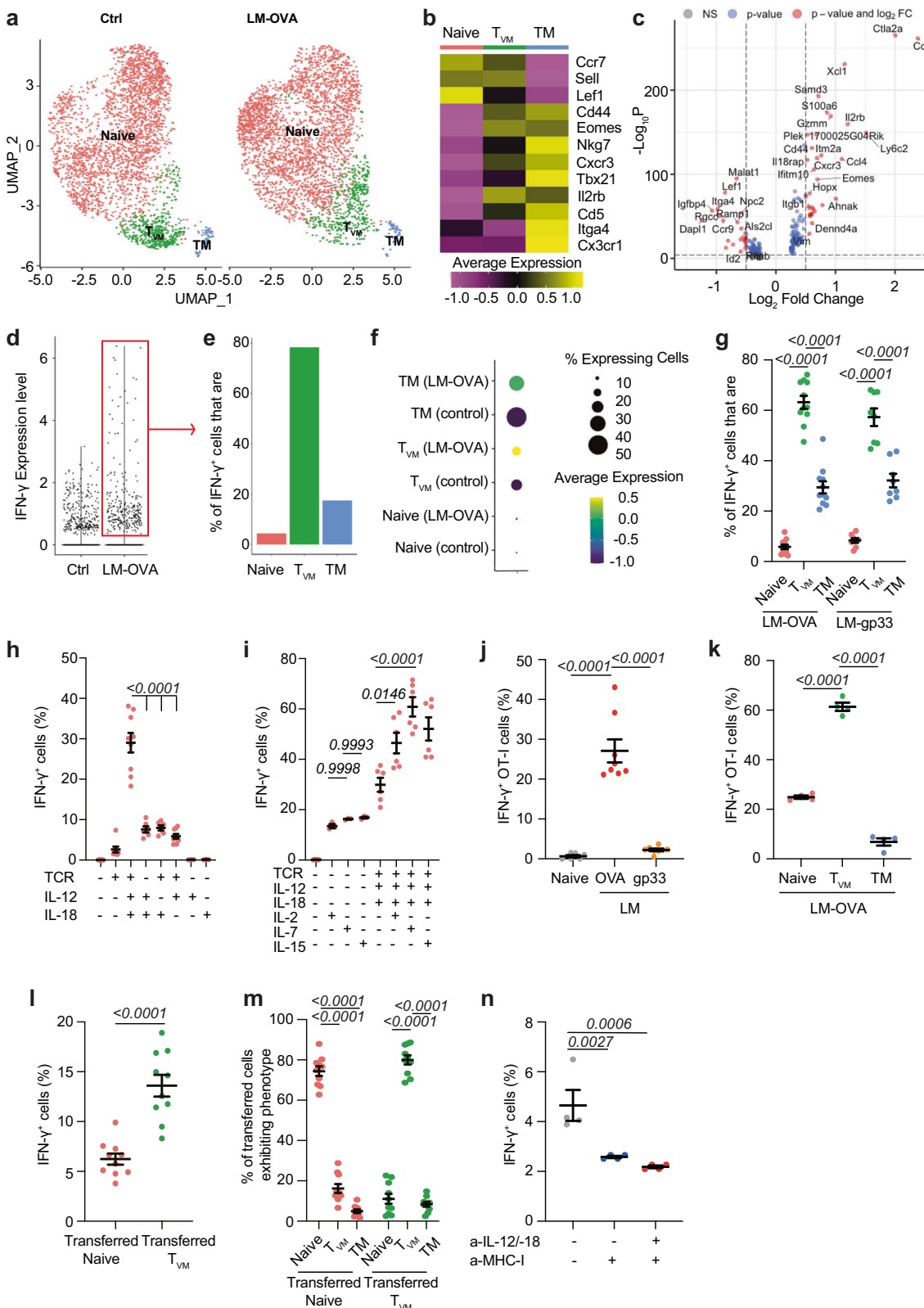

stimulation") and/or inflammatory cytokines, and IFN-γ expression was assessed after 24 h. While IL-12 and IL-18 treatment induced some level of IFN-γ production, adding TCR priming synergized with IL-12/18 (Fig. 4h). IL-2, IL-7, and IL-15, which share the same γ-chain receptor, further synergized with TCR and IL-12/18 stimulation to increase IFN-γ (Fig. 4i). Other known inducers of IFN-γ production did not lead to IFN-γ production by CD8⁺ T-cells (*Fig.S6a*). This suggests that maximum

IFN-γ production by $T_{VM}$ required TCR priming. We validated this in vivo. We crossed GREAT mice with OT-I mice to generate GREAT OT-I T-cells, which were transferred in WT recipients. Mice were then infected with either LM-OVA or LM-gp33. For this, we increased the frequency of OT-I cells transferred to reliably detect OT-I $T_{VM}$. IFN-γ expression by OT-I T-cells was observed upon infection with LM-OVA but not LM-gp33 (Fig. 4j), and was mainly produced by $T_{VM}$ (Fig. 4k),

**Fig. 4 | $T_{VM}$ are the main CD8$^+$ T-cell source of IFN-γ during priming. a–f** CD8$^+$ T-cells from naïve mice (Ctrl) or LM-OVA-infected mice (LM-OVA) were sorted after 24 h and subjected to scRNA-seq analysis ($n = 3224$ Ctrl, 3495 LM-OVA cells from 3 animals). **a** Graph-based clustering of the assembled clusters. **b** Heatmap shows the average expression of selected markers. **c** Volcano plot shows differentially expressed genes between virtual memory ($T_{VM}$) and other CD8$^+$ T-cells (268 variables). Green dots: genes with log2 (fold-change) value > 0.5 or <−05; blue dots: genes with an adjusted $p$ value < 0.05; red dots: genes with log2 (fold-change) value > 0.5 or <−05 and an adjusted $p$ value < 0.05. Two-tailed Wilcoxon rank sum test. **d** Violin plot shows the expression of IFN-γ in Ctrl and LM-OVA infected mice. **e** Graph shows the relative frequency of naïve, $T_{VM}$ or true memory ($T_M$) CD8$^+$ T-cells within the IFN-γ$^+$ T-cell fraction. **f** Dot plot shows the average expression of IFN-γ in naïve, $T_{VM}$ or $T_M$ cells at steady-state and 24 h post LM-OVA infection. **g** GREAT mice were infected with LM-OVA or LM-gp33. Relative abundance of naïve (red), $T_{VM}$ (green) or $T_M$ (blue) T-cells among IFN-γ$^+$ splenocytes was analyzed by flow cytometry 24 hrs post-infection ($n = 11$ LM-OVA-, 8 LM-gp33-infected animals). **h, i** CD8$^+$ T-cells from GREAT mice were stimulated in vitro with anti-CD3/−28 (TCR), and the indicated cytokines. IFN-γ production was analyzed by flow

cytometry after 24 h ($n = 3$–6 animals). **j, k** WT mice were transferred with $2 \times 10^6$ GREAT OT-I CD8$^+$ T-cells. **j** IFN-γ expression by OT-I was quantified by flow cytometry 24 h post-infection with LM-OVA (red) or LM-gp33 (orange) ($n = 6$ naïve, 8 LM-OVA- and LM-gp-33-infected animals). **k** The relative abundance of naïve (red), $T_{VM}$ (green) or memory $T_M$ (blue) among GREAT OT-I IFN-γ$^+$ splenocytes was analyzed by flow cytometry 24 h post-infection (n = 4 animals). **l, m** Sorted $T_{VM}$ and naïve OT-I CD8$^+$ T-cells were transferred into WT hosts. **l** Mice were treated with BFA 6hrs before harvest. IFN-γ expression of naïve (red) and $T_{VM}$ (green) OT-I T-cells among splenocytes 24h post-LM-OVA infection ($n = 9$ for naïve transfer, 10 animals for $T_{VM}$ transfer). **m** Relative abundance of naïve, $T_{VM}$ or $T_M$ CD8$^+$ T-cells among naïve (red) or $T_{VM}$ (green) OT-I IFN-γ$^+$ splenocytes ($n = 10$ animals), analysed by flow cytometry. **n** GREAT mice were infected with LM-OVA and treated with anti-MHC class-I, and anti-IL-12/−18 as indicated. IFN-γ production was quantified in CD8$^+$ T-cells 24h post-infection ($n = 4$ animals). Data are from 1 (**a–f**) or ≥2 (**g–n**) independent experiments. One-way ANOVA with Tukey's multiple comparison test (**h–l**, **n**) and Two-way ANOVA and Šidák's multiple comparison test (**g**, **n**, **m**). Bars indicate the mean ± s.e.m.

confirming the requirement of TCR priming for early IFN-γ induction in vivo. Because the proportion of OT-I $T_{VM}$ was similar in naïve and infected mice (Fig. S6b), we presumed that the phenotype of those cells was stable at the inception of the infection and truly represented $T_{VM}$. To formally prove that $T_{VM}$ had an increased capacity to produce IFN-γ, we isolated naïve OT-I and $T_{VM}$ OT-I, and transferred them into mice that were subsequently infected by LM-OVA. Quantification of IFN-γ production after 24 hours by flow cytometry confirmed that $T_{VM}$ had an enhanced capacity to produce IFN-γ (Fig. 4l), and retained their phenotype at this time point (Fig. 4m). We confirmed that IFN-γ production by CD8$^+$ T-cells required TCR priming in vivo by using Nur77-reporter mice, for which GFP expression is controlled by the Nur77 promoter and correlates with TCR priming[43]. Mice were infected with LM-OVA and after 24 h, we observed a significant shift in GFP expression in IFN-γ$^+$ CD8$^+$ T-cells compared to non-producers (Fig. S6c). Finally, blocking MHC class-I 24 hours post-infection also significantly decreased IFN-γ expression by CD8$^+$ T-cells (Fig. 4n).

Given the propensity of $T_{VM}$ to produce IFN-γ during priming, and the role of IFN-γ in decreasing the avidity of the primary response leading to sub-optimal immunity, we investigated whether the presence of $T_{VM}$ correlated with disease severity in humans. To do this, we made use of the COMBAT dataset, a blood atlas delineating innate and adaptive immune dysregulation in COVID-19[44]. In humans, $T_{VM}$ corresponds to a subset of CD8$^+$ T effector memory RA ($T_{EMRA}$) cells that expresses multiple killer Ig-like receptors (KIRs) and NKG2A/E[45]. We therefore focused on the $T_{EMRA}$ clusters (Fig. S6d) to identify $T_{VM}$. Differentially expressed genes revealed that clusters 2, 3, 5 and 6 are characterized by multiple KIR and NKG2 expression (Supplementary data 1) and were therefore labelled as $T_{VM}$ (Fig. S6e). Analysis of the relationship between $T_{VM}$ and disease severity revealed that enhanced frequency of $T_{VM}$ is associated with severe COVID disease (Fig. S6f). While severe disease, characterized by heightened inflammation, could drive elevated $T_{VM}$ numbers, it also correlates with our finding that CD8$^+$ T-cells mainly sense IFN-γ produced by $T_{VM}$, leading to decreased avidity of the primary response and a subsequently curtailed response during infection. Similar data was observed when we focused on the $T_{EMRA}$ clusters that express markers of $T_{VM}$ but not CD57 (*B3GAT1*) (clusters 2 and 5) to avoid including senescent T-cells in our analysis.

Altogether, our data demonstrate that paracrine IFN-γ sensing by CD8$^+$ T-cell decreases the avidity of the primary response. Because $T_{VM}$ is the subset producing IFN-γ during priming, we speculate that $T_{VM}$ cells regulate the avidity of the CD8$^+$ T-cell response.

## IFN-γ-sensing by CD8+ T-cells does not regulate CD8+ T-cell differentiation and TCR diversity

Our data so far demonstrate that IFN-γ sensing by CD8$^+$ T-cells during priming lowers the avidity of the effector response, thereby limiting

T-cell responses against intracellular pathogens. However, it was unclear whether IFN-γ intrinsically affected T-cell differentiation and fitness, or rather coordinated avidity, expansion, and differentiation. To address this question, we performed single-cell RNA- and TCR-sequencing on OVA-specific CD8$^+$ T-cells from CD8-IFN-γR$^{KO}$ and control mice 9 days post-infection with LM-OVA. Using unsupervised hierarchical clustering, we identified 7 clusters (Fig. S7a), which we manually labelled based on known markers and computed signatures from publicly available datasets of effector and memory subsets (Fig. S7b-d). We distinguished 2 memory subsets based on *Cx3cr1* expression, *Cx3cr1*$^{neg}$ being the least differentiated[46], one effector population and one highly cycling cluster (Fig.5a, b). Control and CD8-IFN-γR$^{KO}$ clusters had similar gene expression profiles, showing that IFN-γ does not affect intrinsic effector or memory potential (Fig. 5c). This agrees with the fact that control and CD8-IFN-γR$^{KO}$ T-cells exhibited similar cytotoxicity (Fig. 1e–h). We also observed that the IFN-γ signature of control cells at the peak of infection was low and not decreased by IFN-γR deletion (Fig. S7e), consistent with our finding that CD8$^+$ T-cells sense IFN-γ during priming rather than during the effector stage.

We then tested whether IFN-γ sensing would alter the differentiation of endogenous CD8$^+$ T-cells. We did not detect any significant differences by flow cytometry, where the proportion of SLECs (KLRGI$^{hi}$CD127$^{lo}$) (Fig. 5d) and MPECs (KLRGI$^{lo}$CD127$^{hi}$) (Fig. 5e) was identical between CD8-IFN-γR$^{KO}$ and control endogenous OVA-specific CD8$^+$ T-cells at the peak of the response. It demonstrated that IFN-γ sensing does not regulate overall CD8$^+$ T-cell differentiation. It was however worth noting that blocking overall IFN-γ during priming increased SLEC proportion (Fig. S7f) and conversely decreased MPEC proportion (Fig. S7g), which might be explained by the function of IFN-γ on other cell types, indirectly affecting CD8$^+$ T-cell differentiation. For example, in models of vaccination, IFN-γ signaling in CD11b$^+$ cells regulates CD62L expression on CD8$^+$ T-cells[47].

We then investigated whether the difference in avidity between control and CD8-IFN-γR$^{KO}$ T-cells was related to the use of a different TCR repertoire. Analysis of TCR usage showed that the OVA-specific T-cell repertoire following LM-OVA infection is unique for each mouse, regardless of the genotype (Fig. S7h), suggesting that TCR repertoires emerging after infection are qualitatively different, as observed during CMV infection[48]. The increased avidity induced by IFN-γR deletion may be the result of the expansion and dominance of a few high-affinity T-cell clones, which would result in decreased TCR diversity. However, IFN-γ sensing by T-cells did not regulate TCR diversity (Fig. 5f), indicating that differences in avidity are not solely due to alternate priming of high-affinity T-cells. To get more insight into the function of IFN-γ on the relationship between clonality and differentiation, we compared the TCR clonal overlap between the different subsets. The overlap between SLECs and the memory populations was limited, about 20%,

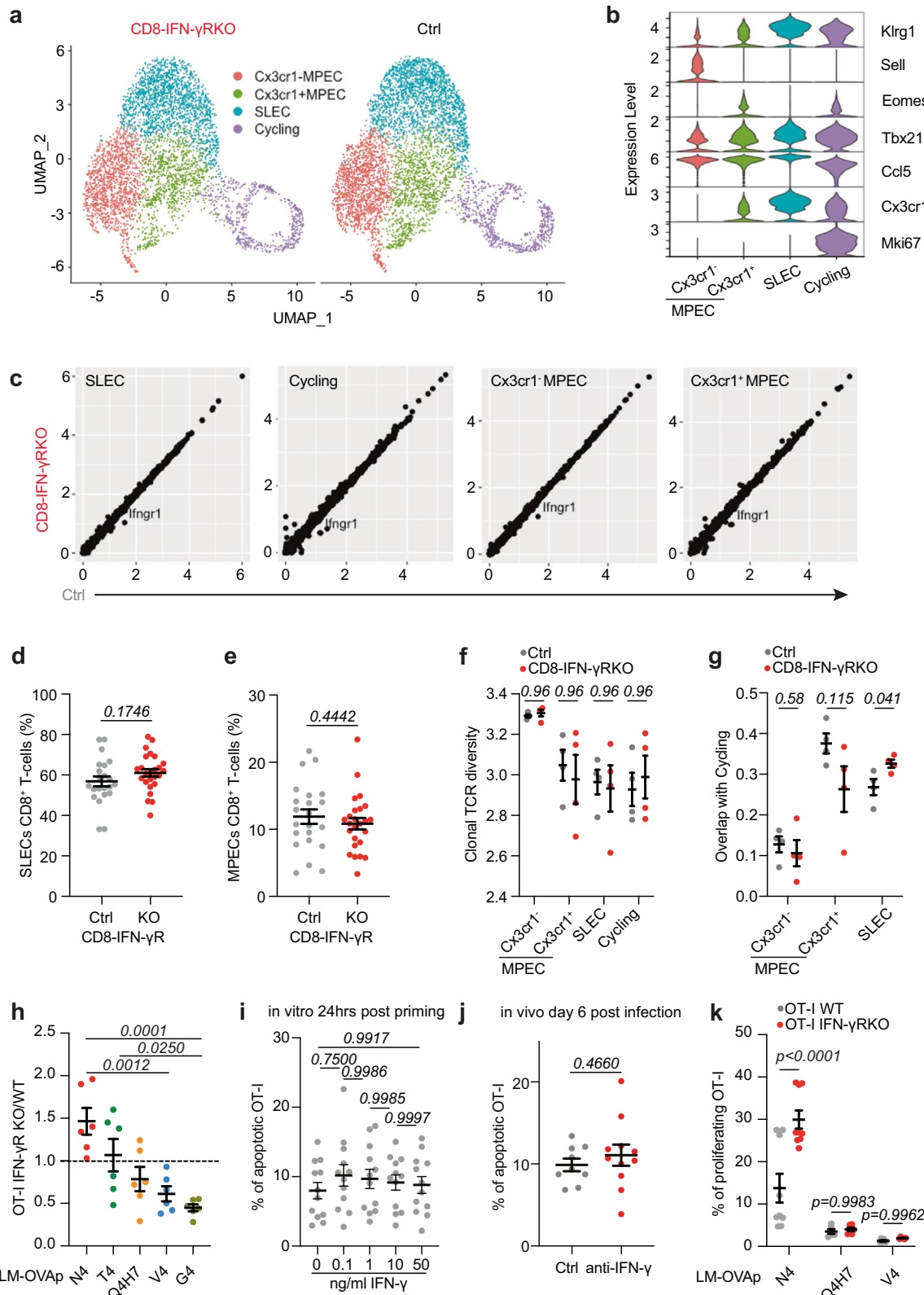

and was not affected by IFN-γR deletion (Fig. S7i). Interestingly, analysis of the overlap between the cycling population and the other subsets revealed that inhibition of IFN-γ sensing enhanced the overlap between cycling and SLEC populations (Fig. 5g), suggesting that the expansion of distinct clones was enhanced following IFN-γR deletion. Similar results were obtained when we focused our analysis on the four most expanded TCRV_β chains (Fig. S8a,b).

Given our data demonstrating that IFN-γR deletion increases the avidity of the effector response, we hypothesized that the SLEC clones exhibiting a high cycling signature following IFN-γR deletion were of high-affinity for their cognate peptide, suggesting that IFN-γ inhibits the expansion of high- but not low-affinity T-cells. To test this, we fixed the CD8+ T-cell clonotype by using the OVA-specific CD8+ T-cell clone OT-I, and varied the affinity of TCR priming. We transferred the

**Fig. 5 | IFN-γR deletion in CD8⁺ T-cells does not regulate overall differentiation and TCR diversity.** **a**–**g** CD8-IFN-γR$^{KO}$ (KO, red) and control (Ctrl, grey) mice were infected with LM-OVA. Splenocytes were isolated 9 days post-infection. **a**–**c**, **f**, **g** N4-tetramer⁺ CD8⁺ T-cells were subjected to scRNA- and scTCR-seq ($n = 5646$ Ctrl, 4837 KO cells from 4 animals). **a** Graph-based clustering of the assembled differentiation states. **b** Violin plot shows the expression of selected markers. **c** Plot shows the average gene expression of Ctrl and KO clusters. Relative abundance of Ctrl and KO SLEC (**d**) and MPEC (**e**) cells($n = 22$ Ctrl, 25 KO animals). **f** Relative TCR clonal diversity (Shannon index) by cell state and genotype extracted from scTCR-seq analysis ($n = 4$ animals). **g** Clonal overlap between the cycling cluster and the other clusters from scTCR-seq analysis ($n = 4$ animals). **h** WT mice were transferred with 50.000 ad-mixed WT and IFN-γR$^{KO}$ OT-I T-cells and infected with LM expressing the indicated OVA peptides. OT-I T-cells expansion was analyzed by flow cytometry after 9 days. Data are expressed as the ratio between IFN-γR$^{KO}$ and WT OT-I cell numbers ($n = 6$ animals). **i** OT-I T-cells were activated in vitro with 50 ng/mL N4 in the presence of increasing doses of IFN-γ. After 36 h, cell apoptosis was assessed by Annexin V staining and quantified by flow cytometry ($n = 12$ animals). **j** WT mice were transferred with OT-I T-cells, infected with LM-OVA and treated with anti-IFN-γ after 24 h. OT-I T-cell apoptosis was analyzed by Annexin V staining 6 days post-infection ($n = 9$ ctrl, 11 anti-IFN-γ-treated animals). **k** WT mice were transferred with 50.000 ad-mixed WT and IFN-γR$^{KO}$ OT-I T-cells and infected with LM expressing the indicated OVA peptides. EdU was injected on day 6 and proliferation of OT-I CD8⁺ T-cells was analyzed 16 h later by flow cytometry ($n = 9$ animals). Data are from 1 (**a**–**c**, **f**, **g**), ≥ 2 (**h**, **k**) or 6 (**d**, **e**) independent experiments. Comparison between groups was calculated using the two-tailed unpaired Student's t-test (**d**, **e**, **j**), One-way ANOVA with Tukey's multiple comparison test (**f**, **g**, **k**) and a Two-way ANOVA and Šidák's multiple comparison test (**h**, **i**). Error bars indicate the mean ± s.e.m.

smallest number of OT-I T-cells allowing reliable detection following priming with peptides exhibiting low-affinity for the OT-I TCR. WT and IFN-γR$^{KO}$ OT-I were co-transferred in mice that were subsequently challenged with LM expressing either the high-affinity OVA peptide N4 or altered peptides of lower affinity. We then compared the expansion of WT and IFN-γR$^{KO}$ OT-I T-cells upon priming. As hypothesized, expansion of IFN-γR$^{KO}$ OT-I T-cells was increased following high-affinity (N4 peptide) priming at the peak of the response compared to WT. Surprisingly, priming with low-affinity peptides led to decreased frequency of IFN-γR$^{KO}$ over WT OT-I T-cells (Fig.5h, S8c), demonstrating that direct sensing of IFN-γ by CD8⁺ T-cells differentially affected T-cell expansion according to TCR avidity, favoring the participation of low-affinity T-cells while restraining the expansion of high-affinity T-cells. This is consistent with the transcriptomics data and our finding that IFN-γ-sensing by CD8⁺ T-cells decreases the avidity of the primary response. Interestingly, IFN-γR$^{KO}$ OT-I T-cells were able to partially rescue the weight loss of WT mice following Influenza-OVA infection (Fig. S8d), most likely because of the enhanced expansion of IFN-γR$^{KO}$ over WT OT-I T-cells when primed with high-avidity.

Because IFN-γ has been shown to induce cell apoptosis[49], we tested whether the restricted expansion of high-avidity T-cells was due to IFN-γ-induced apoptosis. In vitro apoptosis, quantified by Annexin V staining, was not increased by IFN-γ treatment during priming (Fig. 5i). We also tested whether IFN-γ induced apoptosis of high-avidity T-cells in vivo. To do so, OT-I-bearing mice were infected with LM-OVA, and treated with anti-IFN-γ during priming, which inhibits OT-I expansion[23]. Apoptosis was not increased at the peak of the effector response in vivo following anti-IFN-γ treatment (Fig. 5j), confirming that IFN-γ does not affect apoptosis of high-avidity T-cells during LM infection. Our sequencing data rather suggested that T-cell proliferation was restricted by IFN-γ (Fig. 5g and S8b). To test this, WT and IFN-γR$^{KO}$ OT-I were ad mixed and transferred in mice, which were then infected with LM expressing either the high-avidity OVA peptide N4 or altered peptides of lower avidity. OT-I proliferation was quantified by EdU incorporation at the peak of the effector response. IFN-γR deletion enhanced OT-I proliferation when they were primed with high-avidity. Interestingly, this was not observed when OT-I were primed with lower-avidity peptides (Fig. 5k). This overall showed that IFN-γ selectively inhibits the proliferation of high-avidity CD8⁺ T-cells.

We concluded that IFN-γ sensing does not affect T-cell differentiation per se and TCR diversity, but combined analysis of scRNA- and scTCR-seq suggests that IFN-γ signaling in CD8⁺ T-cells specifically curtails the proliferation of high-affinity effector T-cells, promoting expansion of lower-affinity T-cells.

## IFN-γ-sensing by CD8+ T-cells increases the avidity of the memory response

Because low- and high-affinity T-cells are known to have distinct differentiation skewing[50] and IFN-γ sensing by CD8⁺ T-cells lowers the avidity of the primary response, we hypothesized that IFN-γ may rather regulate the relationship between T-cell avidity and differentiation. To investigate this relationship, we first analyzed the differentiation state of adoptively transferred WT and IFN-γR$^{KO}$ OT-I T-cells upon infection with LM expressing altered OVA peptides. IFN-γR deletion had limited effect on the differentiation of T-cells displaying high-avidity, whereas it increased MPEC skewing of T-cells with low-avidity (Fig. 6a, b). This suggested that IFN-γR deletion enhanced memory formation of low-affinity T-cells. To test this, a small number of WT and IFN-γR$^{KO}$ OT-I T-cells were ad-mixed, and transferred in WT mice, which were challenged with LM expressing altered OVA peptides with varying affinities. After 60 days, mice were re-challenged with LM-OVA to analyze OTI expansion. Before re-challenge, the ratio between WT and IFN-γR$^{KO}$ OT-I T-cells was similar to the one observed at the peak of the response for high-avidity priming (Fig. 6c). For OT-I cells primed with lower-avidity, however, the number of WT and IFN-γR$^{KO}$ OT-I T-cells equalized, suggesting that IFN-γR$^{KO}$ OT-I T-cells might preferentially differentiate towards memory (Fig. 6c). This was further accentuated after recall, as expansion of IFN-γR$^{KO}$ OT-I T-cells primed with low-affinity peptides was increased compared to WT during recall responses (Fig. 6d). Priming with high-affinity peptides, however, led to a decreased frequency of IFN-γR$^{KO}$ over WT OT-I T-cells during recall responses as compared to before recall, as already suggested[23] (Fig. 6d). This suggested that IFN-γ sensing by CD8⁺ T-cells increased the avidity of the memory response by limiting the inherent skewing in memory differentiation of T-cells that exhibit lower avidity towards their cognate peptide.

To test this hypothesis, we first analyzed the avidity of OVA-specific SLEC and MPEC CD8⁺ T-cells at the peak of the response. As expected, CD8-IFN-γR$^{KO}$ SLECs were skewed towards higher tetramer binding compared with control (Fig. 6e, f). However, CD8-IFN-γR$^{KO}$ MPECs displayed overall highly variable avidity at this time point (Fig. 6 e, g). To specifically address whether IFN-γ affected the avidity of the memory response, LM-OVA infected CD8-IFN-γR$^{KO}$ and control mice were re-challenged after 60 days, and the expanded memory T-cell population was analyzed. CD8-IFN-γR$^{KO}$ T-cells exhibited lower tetramer binding (Fig. 6h), indicating that the memory compartment was skewed towards lower avidity in CD8-IFN-γR$^{KO}$ mice. Similar results were observed following Influenza infection. We challenged control and CD8-IFN-γR$^{KO}$ mice with the Influenza strain X31-OVA. After 60 days, mice were re-challenged with another virus strain, PR8-OVA. This strategy was used because X31 and PR8 express the same NP$_{68}$ T-cell epitope but different B-cell epitopes, ensuring the virus was not rapidly cleared by memory B-cells. Endogenous OVA-specific T-cells were indeed of lower avidity (Fig. 6i) in the lung, but we could detect very few of those cells. We therefore also analyzed the avidity of the T-cell repertoire towards the dominant flu antigen NP$_{68}$. Both in draining lymph nodes and in the lung, NP$_{68}$ reactive CD8⁺ T-cells expanded less (Fig. 6J, K) and were of lower avidity (Fig. 6L, M) in CD8-IFN-γR$^{KO}$ mice compared to control mice. Altogether, we concluded that IFN-γ improves memory responses.

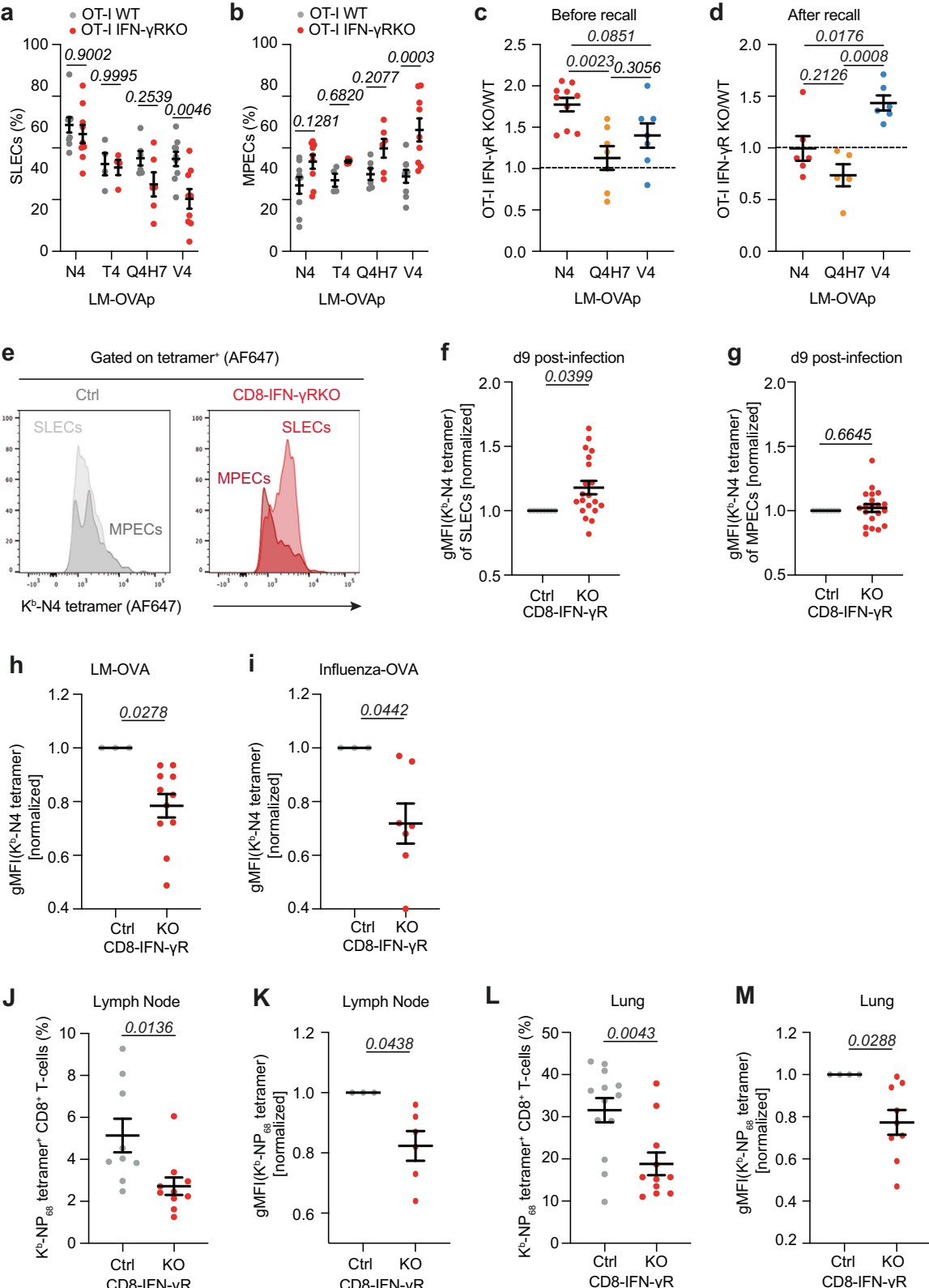

Overall, our data demonstrate that IFN-γ sensing by CD8+ T-cells decreases the avidity of the primary response, leading to sub-optimum control of infection. This is accompanied by increased avidity of the secondary response. T_VM are the main subset providing IFN-γ to other T-cells, linking innate signals to the regulation of CD8+ T-cell avidity, ensuring that the effector and memory pools are composed of T-cell clones displaying a breadth of avidities.

## Discussion

In this study, we described a critical role of IFN-γ as a paracrine modulator of CD8+ T-cell responses by coordinating T-cell expansion, differentiation, and avidity. Individual CD8+ T-cell clones exhibit a remarkable degree of functional and phenotypic plasticity, a prerequisite for effective adaptive responses[3,51,52]. However, such malleability necessitates regulatory mechanisms ensuring robustness at the

**Fig. 6 | IFN-γR deletion in CD8$^±$ T-cells lowers the avidity of CD8$^±$ T-cell memory responses. a–d** WT mice were transferred with 50.000 ad-mixed WT and IFN-γR$^{KO}$ OT-I T-cells, infected with LM expressing the indicated OVA peptides. Relative abundance of OT-I SLEC (**a**) and MPEC (**b**) cells, analyzed 9 days post-infection by flow cytometry (n = 4 for T4, 6–9 animals for other peptides). **c** Relative abundance of transferred OT-I T-cells ≥60 days post-infection (n = 10 for N4, 7 animals for other peptides). **d** Mice were re-challenged after ≥60 days with LM-OVA and OT-I T-cell expansion was analyzed 5 days later by flow cytometry (n = 5 for Q4H7, 6 animals for other peptides). Data are expressed as the ratio between IFN-γR$^{KO}$ and WT OT-I cell number. **e–h** CD8-IFN-γR$^{KO}$ (KO, red) and control (Ctrl, grey) mice were infected with LM-OVA. **e–g** Splenocytes were isolated after 9 days. **e** Representative histograms of N4-tetramer staining within the SLEC and MPEC populations. Normalized N4-tetramer-staining of SLEC (**f**) and MPEC cells (**g**) (n = 20 animals). (**h**) Mice were re-challenged 60 days post-infection with LM-OVA and normalized N4-tetramer

gMFI was analyzed by flow cytometry after 5 days (n = 11 animals). **i–M** KO (red) and Ctrl (grey) mice were infected with X31-OVA. Mice were re-challenged after ≥60 days with PR8-OVA. Tissues were harvested after 5-7 days and stained for CD8$^+$ T-cells and the indicated tetramer. **i** Graph shows normalized N4-tetramer gMFI of N4-tetramer$^+$ CD8$^+$ T-cells in the lung (n = 7 animals). Graph shows the percentage (n = 9 Ctrl, 10 KO animals) (**J**) and normalized NP$_{68}$-tetramer gMFI (n = 6 animals) (**K**) of NP$_{68}$-tetramer$^+$ CD8$^+$ T-cells in draining LNs. Graph shows the percentage (n = 11 Ctrl, 13 KO animals) (**l**) and normalized NP$_{68}$-tetramer gMFI (n = 8 animals) (**M**) of NP$_{68}$-tetramer$^+$ CD8$^+$ T-cells in the lung. KO tetramer gMFIs samples were normalized to the mean Ctrl tetramer gMFI within each independent experiment. Data are from ≥2 (**a–d**), 3 (**h–M**) or 6 (**e–g**) independent experiments. Two-tailed unpaired Student's t-test (**f–M**). One-way ANOVA with Tukey's multiple comparison test (**c, d**) and two-way ANOVA with Šidák's multiple comparison test (**a, b**). Error bars indicate the mean ± s.e.m.

population-wide level. Our data demonstrate that IFN-γ mediates such regulatory mechanisms, not only during infection but also during anti-tumor immunity[53].

Several studies have previously demonstrated that contradictory role of IFN-γ on CD8$^+$ T cell proliferation, differentiation, and effector functions[23,27,28,30,54,55]. Some might be explained by the use of full IFN-γR$^{KO}$ mice, where IFN-γ might target other cells that could indirectly affect T-cell function, or a different immunization protocol. But in addition, our data provide further explanation as to why those discrepancies might exist. Indeed, we found that IFN-γ has a different effect on high- versus low-avidity T-cells. As such, studies that used TCR transgenic mice might have only revealed part of the function of IFN-γ. This is evident when we compared the endogenous response versus the TCR transgenic response for example (Fig. 5d, e versus Fig. 6a, b).

Regulating the avidity of the T-cell response is critical. Increased avidity in the same order of magnitude we observed for IFN-γR deletion has been correlated with resistance to viral infection in humans. Increased control of HIV-1 replication by CD8$^+$ T-cells[56] is correlated with increased avidity, and, consequently, HIV controllers display higher functional avidities of Gag-specific and HLA-B-restricted responses than non-controllers[57]. Similar increased avidity has been observed in patients that cleared Hepatitis C compared to chronically infected patients[58]. This emphasizes the importance of understanding the mechanisms regulating the avidity of the T-cell response.

High-affinity T-cells have a competitive advantage over low-affinity T-cells, with ligand affinity determining the frequency of responding cells[59]. Low-affinity T-cells are nevertheless recruited during an immune response[17], resulting in a physiological range of avidities of the endogenous response of around 100-fold[60,61]. Our data demonstrate that this is in part due to IFN-γ, improving low-avidity T-cell expansion and thereby lowering the threshold for low-affinity T-cells to enter the effector response, overcoming the selective advantage of high-affinity T-cells. Recruiting lower-affinity T-cells and lowering the avidity of primary responses is important in some models to avoid immunopathology driven by high-avidity T-cells[62]. In addition, they become dysfunctional and exhausted in chronic conditions[63]. Interestingly, in metastatic melanoma patients, PD-1$^{neg}$ T-cell clones naturally present within an endogenous repertoire exhibit lower avidity TCRs than PD-1$^+$ T-cell clones, exemplifying the tight relationship between T-cell avidity and exhaustion[64].

Lowering the avidity of the primary response may also be necessary to increase the avidity of the secondary response, as shown here, by allowing T-cell clones that would recognize infected cell with high avidity, to also enter the memory pool, overcoming their inherent bias to becoming effector. Low-affinity and avidity T-cells are intrinsically skewed to become memory T-cells[50] where they were suggested to be important for memory responses towards mutated pathogens[6,65]. This has been recently confirmed by the group of D. Busch in a study that demonstrated that low-avidity T-cells get recruited and expand as long

as they pass a clear TCR avidity threshold. This results in the recruitment of a broad polyclonal repertoire where the presence of low-avidity T-cells allow for flexible secondary responses to mutant epitopes[66]. In this context, IFN-γ regulates the other end of the spectrum by restricting the expansion of high-avidity T-cells, which, as a result, ensures that the avidity of the T-cell response is conserved and high enough during memory stages to enable efficient long-term responses towards native but also mutated pathogens, at which point T-cells that were initially recruited as low-avidity can become major responders.

Unlike for the B cell receptor, TCR cannot undergo affinity maturation. Nevertheless, antigen-primed T-cells significantly increase their antigen responsiveness compared to naïve T-cells in a process called functional avidity maturation. This is traditionally associated with increased co-stimulatory molecules and integrin expression necessary to interact with antigen-presenting cells, which will affect all activating T-cells indiscriminately[67]. Our work shows that IFN-γ sensing by T-cells restrains T-cell avidity through distinct pathways, as co-stimulatory molecules were not affected by IFN-γR deletion (Fig. 5). Mechanistically, we found that IFN-γ is paracrine, mainly provided by T$_{VM}$ to other CD8 T-cells during priming, serving as "Signal 3" to regulate T-cell expansion according to TCR affinity. How IFN-γ signaling regulates clonal selection over the effector and memory pool is unclear. IFN-γ induces the expression of transcription factors known to regulate the threshold of T-cell activation, proliferation, and differentiation, such as c-Myc and T-bet[68,69]. Alternately, IFN-γ induces ICAM-1 expression, which has been shown to regulate memory formation[22,70]. Regulation of ICAM-1 expression in CD8$^+$ T-cells could alter the nature of T-T communication and associated quorum sensing[21]. It is tempting to speculate that those factors may integrate with TCR signaling, but more studies are necessary to elucidate the mechanisms enabling this coordination.

Taken together, our work provides evidence that IFN-γ serve an important autoregulatory role by coordinating CD8$^+$ T-cell avidity and differentiation, decreasing the avidity of the effector response while increasing the avidity of the memory response, ensuring a balanced control of short- and long-term immunity.

## Methods
### Mice
Mice were bred and maintained in the University of Oxford specific pathogen-free (SPF) animal facilities. Mice were routinely screened for the absence of pathogens and were kept in individually ventilated cages with environmental enrichment at 20–24 °C, 45–65% humidity with a 12 h light/dark cycle (7 am–7 pm) with half an hour dawn and dusk period. *CD8a*-Cre$^{GFP}$ (JAX stock no.: 008766), IFN-γR$^{flox/flox}$ (JAX stock no.: 025394), IFN-γ-GREAT-YFP (JAX stock no.: 017580), IFN-γ$^{KO}$ (JAX stock no.: 002287), Nur77-GFP (JAX stock no.: 018974), *CD8a*KO (JAX stock no.: 002665) and OT-I (JAX stock no.: 003831) were purchased from Jackson Laboratory. OT-I mice were bred with CD45.1

mice (The Jackson Laboratory – 002014) to generate congenically marked OT-I CD45.1 cells. C57BL/6 J mice were purchased from Charles River, UK (JAX stock number: 000664). To generate CD8 IFN-$\gamma$R$^{KO}$ mice *CD8a*-Cre$^{GFP}$ mice were crossed with IFN-$\gamma$R$^{flox/flox}$ mice, which were subsequently crossed to Rosa26-tdTomato mice (kind gift from Tal Arnon). All experiments involving mice were conducted in agreement with the United Kingdom Animal (Scientific Procedures) Act of 1986 and performed in accordance to approved experimental procedures by the Home Office and the Local Ethics Reviews Committee (University of Oxford).

## Cell isolation

Spleens were harvested from mice at indicated time points, mashed in 1x PBS and filtered through 70 $\mu$M filter. Splenocytes were resuspended in red lysis buffer (155 mM NH4Cl, 12 mM NaHCO3 and 0.1 mM EDTA in ddH$_2$O) and incubated on ice for 5 minutes, before being washed in PBS twice.

For T-cell isolation, WT or OT-I CD8$^+$ T-cells were isolated through negative separation from lymph nodes and spleens of 6- to 12-week-old WT or IFN-$\gamma$R$^{KO}$ OT-I mice, using the MojoSort$^{TM}$ CD8$^+$ T-cell isolation kit and magnets (Biolegend, #480008 and #480019). Isolated T-cells were resuspended in complete RPMI (RPMI 1640 [Gibco, #21870-076] supplemented with 2% FCS and 100x Penicillin-Streptomycin [Gibco, #10378-016]).

## Infection and treatments

Mice were given an intravenous (i.v.) injection of 20*10$^3$ colony-forming units (cfu) of LM expressing either a secreted form of OVA (LM-OVA) or one of the LM following strains expressing the indicated peptides (LM-N4, LM-T4, LM-Q4H7, LM-V4, LM-G4 or LM-gp33)[17,71]. LM strains were provided by Dietmar Zehn (TU Munich) except LM-gp33, which was purchased from Nanjing Sungyee Biotech. Frozen down LM aliquots were expanded in Brain Heart Infusion (BHI) broth (Sigma, #53286-100G) and LM suspensions were plated on BHI agar plates (Sigma, #70138-500G). LM (200.000 cfu for 24 h experiments; 20.000 cfu for d7-10 experiments) was injected intravenously when they were in exponential phase of growth. For memory responses, mice were rechallenged at least 60 days after the primary infection with 200.000 cfu.

In some experiments, OT-I T-cells (2–3 × 10$^6$ cells or 50.000 cells for 24 h or d7-10 experiments, respectively) were transferred into mice recipient by intravenous injection the days before infection.

In some experiments, mice received a single intraperitoneal (i.p.) injection 16–24 hours post infection of 75 $\mu$g of isotype matched control antibody (rat IgG1, BioXCell, #BE0088) or anti-Interferon-gamma (BioXCell, clone: XMG1.2). In some experiments 50 ug of isotype matched control antibody (rat IgG2a, BioXCell, #BE0089) or anti-NK1.1 (BioXCell, clone: PK136) was injected intraperitoneally two days and one day prior to infection, as well as on day 4 and 6 post-infection.

For subsequent staining of intracellular cytokines, mice were injected i.p. with 250 $\mu$g BFA 6 hours before being killed.

For primary infection with Influenza virus, mice weighing >20 g were anaesthetized using isoflurane and intranasally administered with 4 × 10$^4$ or 4×10$^5$ PFU of X31-OVA influenza A virus in PBS. Mice were weighted for 14d following infection. For memory responses, mice were rechallenged at least 60 days after the primary infection with 10$^6$ PR8-OVA. X31-OVA and PR8-OVA were provided by Paul Thomas (St. Jude Children's Research Hospital).

## Quantitative real-time PCR

Lungs were removed and roughly dissected before digestion in 1 mg/mL collagenase D (Roche) and 10 $\mu$g/mL DNaseI in RPMI for 45 min at 37 °C. Tissue was homogenized using the soft tissue homogenizing CK14 kit (Precellys, Stretton Scientific, #03961) in 300 $\mu$L RLT lysis buffer (Qiagen). RNA was isolated using the Qiagen RNeasy Micro kit with a DNA digestion step (Qiagen). Normalized amounts of isolated RNA were reverse transcribed using a high-capacity reverse transcription kit (Applied Biosystems, #436881). Quantitative real-time PCR (qPCR) was performed using premade, Taqman probes (Life-Technologies) and run on an Applied Biosystems ViiA 7 Real-Time PCR system. Gene expression values, relative to the housekeeping gene(s) as indicated, were calculated using the 2$^{\Delta ct}$ method.

## In vitro Activation and Treatment

For in vitro re-stimulation experiments, splenocytes were activated 7–10 days after infection with different concentrations of N4 peptide or phorbol 12-myristate 13-acetate (PMA) (2 ng/mL) and ionomycin (20 ng/mL).

For in vitro stimulation experiments, naïve T-cells were seeded in 96-well U-bottom together with bone marrow-derived dendritic cells (BMDCs) loaded with the indicated OVA peptide (Proteogenix) at 10 ng/mL or antibodies. For experimental conditions including TCR stimulation, wells were pre-coated with anti- CD3$\epsilon$ (Biolegend, #145-2C11) prepared in PBS at 1 $\mu$g/mL for 2–3 hours at 37 °C before being emptied for seeding. For ICAM-1 stimulation, wells were pre-coated with 5 $\mu$g/mL ICAM-1 (Biolegend, #553006) for 2–3 hours at 37 °C before being emptied for seeding. Following coating with anti-CD3$\epsilon$ and/or ICAM-1, cytokine preparations were added to the wells according to the specified experimental conditions, which included different combinations of IL-12 (Biolegend, #577004), IL-18 (Biolegend, #767004), IL-15 (Biolegend, #566302), TNF (PeproTech, #315-01A-20uG), IL-33 (Biolegend, #580504), IFN-$\alpha$ (Biolegend, #752804), IFN-$\beta$ (Biolegend, #581304), IL-2 (Biolegend, #575404), IL-1$\beta$ (Biolegend, #575102), and IL-7 (PreproTech, #217-17). For co-stimulation, 1 $\mu$g/mL anti-CD28 (Biolegend, #37.51) was also added to wells pre-coated with anti-CD3$\epsilon$. The cells were cultured overnight under the specified conditions at 37 °C in 5% CO$_2$ before being subjected to staining and flow cytometry analysis.

For subsequent staining of intracellular cytokines, cultured cells were treated with 7 $\mu$g/mL brefeldin A (BFA, ChemCruz #sc-200861A) 30 minutes post-stimulation and incubated for 4.5 hours at 37 °C in 5% CO2 before further processing.

## Generation of BM chimeras

IFN-$\gamma$$^{KO}$ Recipient mice were irradiated with 4.25 Gy per cycle for two irradiation cycles, 4 hours apart. Bone marrow from IFN-$\gamma$$^{KO}$, *CD8a*$^{KO}$ or C57BL/6J mice were prepared from femur and tibia. Cell suspension was filtered through a 40 $\mu$M strainer shortly prior to intravenous injection of 100 $\mu$L (equivalent to 1.25 ×10$^5$ cells/mouse). Mice were kept for two weeks post-transplantation on antibiotics (Enroflocaxin) in drinking water to avoid infection.

## In Vivo Cytotoxicity Assay

Isolated and washed splenocytes were resuspended in complete RPMI and divided into two 15 mL falcon tubes, each containing 1 mL splenocyte suspension. OVA peptide SIINFEKL (N4) was added to one splenocyte suspension to a final concentration of 10 $\mu$g/mL. Both the peptide-treated and untreated splenocytes were incubated at 37 °C for 30 minutes and shaken half-way. The cells in both tubes were counted and resuspended to obtain 1 mL suspensions containing 10*10$^6$ cells. Peptide-loaded and non-loaded cell suspensions were incubated with 1 $\mu$M eFluor 670 and 2 $\mu$M CFSE in 1x PBS, respectively. The dye-stained cell suspensions were centrifuged and resuspended in 1x PBS, before being mixed at a 1:1 ratio for injection into LM-OVA-infected recipient mice, as well as a naïve mouse used as control. Splenocytes were isolated from the spleens of infected and naïve mice 24 h following injection, and cytotoxic killing by OVA-specific CD8$^+$ T-cells was assessed through flow cytometry analysis. The degree of cytotoxicity is given as the percentage of target-cell lysis relative to the naïve mouse, calculated by the following formula: $100-100*(R_{infected}/R_{naïve})$, where

the R values are equal to the % peptide-loaded population/% non-loaded population ratios in the infected and naive mice.

## Flow Cytometry

Single-cell suspensions obtained from spleen or cultured CD8+ T-cells were stained in V-bottom 96-well plates in flow cytometry buffer (2% FCS, 2 mM EDTA, and 0.02% sodium azide in 1x PBS). Live dead staining and surface staining was performed using Zombie NIR Fixable Viability Kit (Biolegend, #423106/423105), TruStain FcX™ (anti-mouse CD16/32, Biolegend, #101319) and fluorochrome-conjugated primary antibodies (Biolegend, Cell Signaling Technology or BD Biosciences), respectively. Apoptosis was analyzed by Annexin V-PE (Thermo Scientific, # A35108) staining. For experiments that included tetramer staining, cells were incubated with either Alexa Fluor 647- or BV421-conjugated, N4-specific and $NP_{68}$-specific MHC I tetramers (obtained from the National Institutes of Health Tetramer Core Facility [Emory University, Atlanta]) diluted 1:500 in flow cytometry buffer for 30 minutes at room temperature, prior to surface staining. If not otherwise indicated, all cells were fixed either in 4% paraformaldehyde (PFA) for 10 minutes at room temperature or using eBioscience™ Foxp3/Transcription Factor Staining Set (Invitrogen, #00-5523-00) for 30 minutes at 4 °C. Intracellular transcription factor staining was performed after fixation and permeabilization using fluorescently labelled primary antibodies. Intracellular cytokine staining was performed after 15 minutes fixation using BD Cytofix/Cytoperm™ Fixation/Permeabilization Kit (BD Biosciences, #554714). Flow cytometry data were recorded on BD LSRII or FortessaX20 using BDFACSDiva (v8.0) software and analyzed using FlowJo™ software (v10.4.2, Tree Star).

Surface markers used for flow cytometry analysis included: anti-CD8 (Biolegend, clone: 53-6.7), anti-CD4 (Biolegend, clone: RM4-5), anti-CD69 (Biolegend, clone: H1.2F3), anti-CD44 (Biolegend, clone: IM7), anti-CD49d (Biolegend, clone: R1-2), anti-NK1.1 (Biolegend, clone: PK137), anti-KLRGI (Biolegend, clone: 2F1/KLRF1). Antibodies used for intracellular cytokine staining included anti-IFN-gamma (Biolegend, clone: XMG1.2).

All antibodies used and dilutions can be found in Supplementary data 3.

## In vivo EdU labeling

In vivo proliferation was analysed using the Click-iT™ Plus EdU Alexa Fluor 647 Flow Cytometry Assay Kit (Thermo Scientific, #C10634) following the kit's instructions. Briefly, mice were given an i.v. injection of 1 mg/mL EdU (prepared in sterile DPBS) 16 h prior to tissue collections. Single-cell suspensions were stained as described previously for surface antigens in V-bottom 96-well plates. Following extracellular staining, cells were fixed with 4% PFA for 15 min at room temperature, before being washed with 1% BSA in PBS and resuspended in 50 μL of 1X Click-iT® saponin-based permeabilization and wash reagent and incubated for 15 min at room temperature. 250 μL of Click-iT® master mix (prepared according to the kit's instructions) were then added to each well and incubated for another 30 min at room temperature. Cells were subsequently washed and analyzed by flow cytometry.

## Cell binding avidity assays

For acoustic force spectroscopy of T-cell avidity using the Z-Movi cell avidity analyser (LUMICKS, The Netherlands), microfluidic chips were functionalised with 1 M NaOH followed coating with poly-L-lysine (Sigma Aldrich). Chips were kept dry in a 37°C incubator, with repeated aspiration of any residual liquid. For target cell monolayer seeding, microfluidic chips were rehydrated with prewarmed media. B16 tumour cells expressing OVA (70 – 80% confluency) were then seeded at a density of 100 million cells/mL and frequent checking under the microscope to ensure no bubble formation and appropriate seeding density. Cells were incubated for 3 h at 37°C, with fresh change of media in between. Experiments were performed with sorted N4-tetramer+ cells isolated from 4 control and 4 CD8-IFN-γR$^{KO}$ mice 9 days after LM-OVA infection on each chip. Sorted cells were cultured for 3 days prior to measuring cell avidity. Effector T-cells were stained with CellTrace Far Red Proliferation Kit (ThermoFisher Scientific) according to manufacturer's instructions. The labelled cells were then seeded at a density of 10 million cells/mL onto the microfluidic chip containing the target cell monolayer and incubated for 15 min prior to increasing force application to measure cellular avidity. The 2 different types of effector cells were evaluated on the same microfluid chip, and the order was randomised between chips on repeated runs. Image-based automated detection of T-cell detachment was performed using the Oceon software (LUMICKS) and analysis was conducted according to manufacturer recommendations.

## Single-cell RNA sequencing

For CD8+ T-cell sequencing during priming: CD8+ T-cells from 3 naïve mice and 3 mice infected with LM-OVA for 24 h were sorted from splenocytes, cryopreserved in 20% FBS and 10% DMSO in RPMI and further processed by Single Cell Discoveries, Netherlands. Samples were further processed in accordance with 10x Genomics single cell protocols. Single cell libraries were prepared using the Chromium 3′v2 platform (10× Genomics, Pleasanton, CA) following the manufacturer's protocol. In brief, single cells were encapsulated into gel beads in emulsions (GEMs) in the GemCode instrument followed by cell lysis and barcoded reverse transcription of RNA, amplification, shearing and 3′ adaptor and sample index attachment. Approximately, 5000 to 7000 cells were recovered. Libraries were sequenced on the Illumina NovaSeq 6000. Read mapping, alignment to GRCm38 and quantitation of sample count matrices was performed with the 10x Genomics Cell Ranger pipeline (v 4.0.0).

For CD8+ T-cell sequencing at the peak of the primary response: N4-tetramer+ CD8+ T-cells from 4 control and 4 CD8 IFN-γR$^{KO}$ mice infected with LM-OVA for 9 days were sorted from splenocytes. Each mouse from each genotype was labelled with TotalSeq™ Hashtags (Biolegend) and mixed. Approximately 20,000 cells per sample were loaded onto the 10X Genomics Chromium Controller (Chip K). Gene expression, feature barcoding and TCR sequencing libraries were prepared using the 10x Genomics Single Cell 5′ Reagent Kits v2 (Dual Index) following manufacturer user guide (CG000330 Rev B). The final libraries were diluted to ~10 nM for storage. The 10 nM library was denatured and further diluted prior to loading on the NovaSeq6000 sequencing platform (Illumina, v1.5 chemistry, 28 bp/98 bp paired end for gene expression and feature barcoding, 150 bp paired end for TCR libraries).

## scRNA-sequencing analysis

Datasets were analyzed in R (v 4.1.3) using Seurat version 4.0.6[72]. For CD8+ T-cell sequencing during priming, we filtered out T-cells having less than 600 and more than 5,000 detected genes, cells in which mitochondrial protein-coding genes represented more than 10% of UMI. Cells were then further filtered based on the expression of *Cd2*, *Cd8a* and *Cd8b1*. Samples were then integrated with the IntegrateData function and normalized with the scTransform function of Seurat and variation associated with mitochondrial and ribosomal UMI percentage were regressed out. Principal components were calculated using the top 3,000 variable features. These genes were used as input for principal component analysis (PCA), and significant PCs ($n = 30$) identified using Seurat ("JackStraw" test and "Elbowplot"). Clustering was performed with the Louvain algorithm ($n = 30$ PCs, resolution = 0.3).

For CD8+ T-cell sequencing at the peak of the primary response, we filtered out T-cells having less than 500 and more than 5,500 detected genes, cells in which mitochondrial protein-coding genes represented more than 5% of UMI and cells in which the percentage of largest genes was more than 15% of UMI. Cells were then further

filtered based on the expression of *Cd2*, *Cd8a* and *Cd8b1*. Samples were then integrated with the IntegrateData function and normalized with the scTransform function of Seurat and variation associated with mitochondrial UMI percentage were regressed out. Principal components were calculated using the top 3,000 variable features. These genes were used as input for principal component analysis (PCA), and significant PCs (n = 20) identified using Seurat ("JackStraw" test and "Elbowplot"). Clustering was performed with the Louvain algorithm (n = 20 PCs, resolution = 0.3). A large cluster corresponding to a contamination with naïve CD8$^+$ T-cells was removed and dataset was re-normalized, scaled and PCA and UMAP were re-calculated.

For differential expression analysis, NormalizeData and ScaleData were run on the RNA assay of the integrated data. Significant differentially expressed genes between clusters were identified using the "FindAllMarkers" function, Wilcoxon test and selecting markers expressed in at least 25% of cells. Significant differentially expressed genes between stimulation within clusters were identified using the "FindMarkers" function, Wilcoxon test and selecting markers expressed in at least 25% of cells. Pathway analysis was performed with Fast gene set enrichment analysis (fgsea), using the Gene Ontology or the Reactome pathway repositories. IFN-γ, effector and memory signatures (Supplementary data 2) were computed using the package UCell (v.1.3)[73]. Cell-cell communication analysis was performed using the package NicheNet[41]. Data was visualized using EnhancedVolcano (v1.12.0), GGplot2 (v3.3.5) and ggpubr (v0.5.0).

For TCR sequencing analysis, paired chain TCR sequences were obtained through targeted amplification of full-length V(D)J segments during library preparation. Sequence assembly and clonotype calling was done through cell ranger's immune profiling pipeline (cellranger multi). TCR profiling on filtered contig annotations was done using R package scRepertoire version 1.1.4[74]. Only cells for which both TCRa and TCRb could be identified were used. Clone calling was done for each sample set independently before integration in the Seurat object. Diversity was done by calculating the Shannon Index for each mouse, using the function clonalDiversity in scRepertoire.

## Statistical analysis

One-way ANOVA was selected for pairwise comparisons across multiple experimental conditions and unpaired student's t-tests were used to compare two conditions for statistical significance. EC$_{50}$ between groups was calculated by fitting a 3-parameter fixed-slope Hill function and confirming good fit by the R square function and visual inspection before performing a F-test to compare the model parameters. Data were considered statistically significant when $p < 0.05$. Data are presented as mean ± SEM. Statistical analysis was performed using Graphpad (V8.4.1, Prism software).

## Reporting summary

Further information on research design is available in the Nature Portfolio Reporting Summary linked to this article.

## Data availability

The mouse scRNAseq and scTCRseq data generated in this study have been deposited in the GEO database under accession code GSE244203. Datasets reused in this study: EGAS00001005493[44]. All data are included in the Supplementary Information or available from the authors upon reasonable requests, as are unique reagents used in this Article. The raw numbers for charts and graphs are available in the Source Data file whenever possible. Source data are provided with this paper.

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

## Acknowledgements

We thank Tal I. Arnon, Michael L. Dustin, Vivian W. C. Lau, Anne Chauveau and Mariana Borsa for critical reading of the manuscript. We thank the Lumicks team, in particular Brittany Wingham, for assistance with cell binding avidity experiments and Jonathan Webber for assistance with cell sorting. We thank Yavuz F. Yazicioglu and Barbora Schonfeldova for helpful discussions. We thank the NIH Tetramer facility for all the tetramers used in this study. This research was funded in whole, or in part, by the UKRI (BBSRC BB/R015651/1 to A.G.), Cancer Research UK (CR-UK) (C5255/A18085 through the Cancer Research UK Oxford Centre and 29549 to A.G); the Kennedy Trust for Rheumatology Research (KENN151607 and KENN202112 to A.G), John Fell Funds (0006162 to A.G), MLSTF funds (to A.G) and Kennedy Studentship (to L.F.K.U). For the purpose of Open Access, the author has applied a CC BY public copyright licence to any Author Accepted Manuscript version arising from this submission.

## Author contributions

L.F.K.U performed all experiments, except as noted thereafter; H.C. conducted the in vitro T cell stimulation experiments in Fig. 4 and assisted with the in vivo cytotoxicity assay; S.L.O and A.J.M helped with Influenza infections; J.N.M and J.M.M assisted with some LM infections and conducted the ELISAs; T.P. and A.J.H. assisted with some experiments; D.L. helped perform the Lumicks z-Movi experiments. A.G. analysed the sequencing data, and contributed to conceptualization, funding acquisition, project administration and supervision. A.G. and L.U. wrote the manuscript. All authors edited the manuscript.

## Competing interests

The authors declare no competing interests.
