## [Peer Review File · Nature Communications]

Reviewers' Comments:

Reviewer #1:

Remarks to the Author:

In this study, the authors demonstrated that IFN-g sensing by CD8+ T cells regulates avidity of the CD8+ T cell population during primary and memory responses. The authors showed that IFN-g limits the expansion of CD8+ T cells with high avidity for their cognate peptide, while the expansion of CD8+ T cells with low avidity is preserved. The authors also demonstrated that virtual memory T (Tvm) cells are major cellular sources of IFN-g and that IFN-g works in a paracrine manner. Finally, they showed that IFN-g sensing by CD8+ T cells increases the avidity during the memory response.

In this study, the authors tried to reveal the direct influence of IFN-g signaling on CD8+ T cells encountered with cognate antigens, especially focusing on how varying CD8+ T cells with a breadth of avidities are selected and expanded throughout primary and memory immune responses. Although they investigated this subject by using various mouse models including IFN-g receptor KO mouse (CD8-IFN-gRKO), further experiments and explanations are required to improve this study.

1. The authors showed that IFN-g sensing by CD8+ T cells decreases avidity during primary responses. One of possibilities is that IFN-g selectively eliminates CD8+ T cells with high avidity by inducing apoptosis or cell death, thereby decreasing avidity of the antigen-specific CD8+ T cell population. Please test this hypothesis by performing further experiments.
2. In Figure 1, please show quantitative kinetics of OVA-specific CD8+ T cells in WT and KO mice and kinetics of viral load.
3. In Figure 1G, please examine granzyme B and perforin in addition to LAMP-1.
4. In the evaluation of avidity by MHC-I tetramer binding, please explain a justification of normalization by the CD3 expression.
5. In NK cell depletion analysis, please measure EC50 for more accurate analysis of avidity.
6. In Figure 4, the UMAP projection split by control mice and infected mice is recommended to be shown.
7. In Figure 4B, the scale axis of the color bar may be opposite (as naïve cells are known to have a high expression of Ccr7, Sell and Lef1).
8. In Figure 4D, as the 10x genomics cell ranger output files are a sparse matrix, obvious criteria for determining IFN-g+ cells needs to be shown (for example, in violin plot or feature scatter plots).
9. In Figure 4M, it will be better to show if Tvm cells in this public dataset express Ifng.
10. In Figure 5B, the CX3CR1+ MPEC still show some expression of effector-related markers like Klrg1 and Tbx21, so the authors should examine additional markers to confirm these population as MPEC population. Additional public dataset may be used for further validation.
11. In Figure 5C, CD8-IFN-gRKO mice still seems to have Ifngr1 expression. Authors need to show the actual expression level of Ifngr1 in CD8-IFN-gRKO mice.
12. Figure legend for Fig. 1 mentions that "Data are pooled from 3 (A-D, G, J, n = 11-13) or 4 (F, H-I) independent experiments. How many mice were used per experiment? What is the n number for Fig. 1F, H-I? n number needs to be more clearly indicated.
13. Several studies have previously demonstrated that KO of IFNGR1 does in fact affect CD8+ T cell proliferation, differentiation, and effector functions (Science 2000, 290(5495):1354-1357; J Exp Med 2005, 201(7):1053-1059; J Immunol 2010, 184(6):2855-2862; J Immunol 2012, 189(2):659-668), though the exact positive or negative impact of IFN-g on CD8+ T cells remains to be conclusively determined. Yet, no such impact on cell expansion or effector functions were reported by the authors as seen in Figure 1. The authors should provide some explanatory discussion of this discrepancy between the findings of this and study others in the manuscript.

Reviewer #2:

Remarks to the Author:

This article uses a combination of mice genetically lacking expression of IFNgR selectively on CD8 T cells, as well as a series of BM chimeras to dissect the impact of IFNg sensing by CD8 T cells on

their avidity. Using the CD8-IFN γ R KO mice they show that a loss of IFN γ sensing by CD8 T cells has relatively little effect on response magnitude but results in an increased TCR affinity (by comparing tetramer staining while controlling for TCR levels) following infection with either influenza or *Listeria monocytogenes*. This also translated to a shift in functional avidity. This was also supported by a nice experiment (Fig 5H), where Lm-OVAp variants were used to show a selective impact of IFN γ RKO on high avidity compared to low avidity cells in the effector response. Using neutralizing Ab experiments the authors showed that early (16-24h) administration of anti-IFN γ Abs also increased the avidity of T cells in the effector response, while later (d5/6) did not, suggesting that IFN γ sensing during priming was important. While this seems reasonable, a caveat here is that neutralizing Ab can persist for up to a week so the critical timing here is unclear.

Through NK depletion and mixed BM chimeras the study found that CD8 production of IFN γ was critical for this effect. The authors showed that the main producers of IFN γ after Lm infection were Tvm cells. Importantly, while the phenotype of the cells (CD44^{hi}, CD49^{lo}, CD122^{hi}) is clearly a Tvm phenotype in unchallenged mice, can the authors be certain these are not early activated cells at 24h after Lm-OVA infection? Do they find a similar frequency of these cells in the control mice?

They showed that Tvm cells produce the largest amounts of IFN γ when stimulated through their TCR in addition to cytokines, however I note that they do not include a condition of combined IL-12 and IL-18 stimulation (standard for stimulating Tvm production of IFN γ) in the absence of TCR stimulation.

The authors then conclude that Tvm cells need to be stimulated through their TCR (rather than in a bystander manner) in order to produce the IFN γ that regulates avidity. But they adoptively transfer total OTI cells from OTI x GREAT mice and show that those cells produce IFN γ upon infection with Lm-OVA. How are they attributing this to Tvm cells? The results in Fig 4I and J simply seem to indicate that they have an antigen-specific response by OTI cells to Lm-OVA. Of course, once stimulated by OVA, the cells would not exhibit a Tvm phenotype anymore as they would be antigen-experienced.

There are many other issues with this study. Generally, the study is underpowered and conclusions have been drawn on relatively limited datasets or on experiments that provide supportive but not definitive results. Details of some other key issues are as follows:

Were the control mice (WT) used in this study IFN- γ R1 flox/flox mice? Given the relatively mild phenotype differences after x31 infection, it is important to demonstrate that this isn't a consequence of minor differences in background. In addition, in my extensive experience, HKx31 strain of flu is only moderately virulent and is never lethal in B6 mice. Is Fig 1D actually showing death or is it showing a humane endpoint and if the latter please clearly articulate what this is in the legend and in the methods. It is important that this distinction is also made in the text. In addition, the legend indicates that mice were infected with 10e5 pfu while the methods indicate that mice were infected with 4 x 10e4 or 4 x 10e5 pfu. Please clarify.

For all figures, the number of times each of these experiments has been performed needs to be included in the figure legends. For example, Fig. 1F, H, and I have only 4 mice per group, and should be repeated to firm up conclusions.

Fig S2 A-C, why does panel A have so few data points when these data are derived from the same experiments as those shown in panels B and C? All data points should be included in A. Similarly, for Fig 3A-C. Why does panel C have so few datapoints when this data comes from the same experiment as those depicted in A and B?

Also, for Fig 2E-J n= 2 or 3. Firstly, in the legend it is stated "each dot represents an experiment". What is meant by this? Why doesn't each dot represent the data from one mouse? Secondly, it is impossible to get a statistically significant value when n<3 so I do not understand how this was obtained for Figs 2F and 2H. Additional datapoints must be included here.

For Fig. S2A, E and F there are too few data points to definitively state difference or no difference, especially if they have been performed only once. If performed more than once, an example of an additional experiment should be included in supplementary data to give confidence that this finding (with a current n of only 3 or 4) is reproducible.

The authors state on top of page 5 "as the increased tetramer binding in CD8-IFN- γ RKO CD8+ T-cells was already present 5 days after infection (Fig.S2G)." This is a dubious call. The datasets look to be of almost identical magnitudes with the exception of one higher value in the KO mice, and this difference is almost certainly not significant. Revise this interpretation.

Fig 2. A-D, what is the gating strategy here? What about lymphocyte and live gating? It should be included in supp data, including representative splenic tetramer staining.

What does it mean that mice were treated with anti-IFN γ Ab at days 5-6? Is this one or two administrations? This should be included in methods.

The cited paper (28: Tewari et al, 2007) for the two waves of IFN γ production during infection seems incorrect. Haring et al, 2005, J Immunol suggests that CD8 T cells are responsive to IFN γ within 12 h after Lm infection. The authors should cite the correct paper or alter their statement on IFN γ production after Lm infection.

For Figure 2A, F and H, and Fig 3H these data points should not be joined by lines. These are not paired analyses and there is nothing connecting individual WT mice with individual KO mice. These data should be shown independently. With regard to Fig 3H, the IFN γ signalling data should not be 'normalized' to the IFN-g KO. This can be used as a background that can be subtracted but fold difference from (theoretical) zero is unachievable. For this reason, the statement at the top of page 6: "Ablation of CD8-induced IFN- γ led to 40% inhibition of IFN- γ signaling (Fig.3H)" may be inaccurate and should be recalculated.

Similarly, the lines in Fig 6A and B should be removed. These are not continuous samples.

The analysis of human cells shown in Fig 4 K-M concludes that elevated frequencies of Tvm cells in humans corresponds to severe COVID. Firstly, how many patient samples is this? The legend indicates for M that each dot is a patient but there are no dots here. Secondly, this could equally indicate that severe disease, characterised by heightened inflammation, drives elevated Tvm numbers, rather than the other way around.

For Fig 6C, why were the mice challenged with Lm-OVA? If the point of this experiment was to determine the extent to which priming with different avidities influenced memory formation, why not just look in the memory pool at d60 after Lm-OVA priming? This would eliminate the confounding influence of differential expansion upon rechallenge.

Minor points

Page 2, last paragraph - "...regulates clonal selection to coordinate avidity....", and "...skewing their differentiation towards effector cells"

The statement in paragraph 4 of page 5 "This demonstrated that the increased avidity observed in CD8-IFN- γ RKO mice was not the result of indirect increased IFN- γ signalling in other cells." is not technically correct. The earlier experiments where IFN γ R was knocked out only on CD8 T cells made this point.

This manuscript should cite the recent paper by Dirk Busch's group - Straub et al (Immunity) 2023 - which clearly demonstrates the proliferative advantage of high avidity T cells, their relative imperviousness to the multitude of low avidity responses, and in particular, the proliferative and protective capacity of low avidity T cells recruited into primary memory, upon rechallenge with a mutant peptide for which they exhibit higher avidity than for the original epitope. In the Discussion, the authors should speculate what the implications of this study are for their current data set.

Reviewer #3:

Remarks to the Author:

The authors explore the role of IFN- γ sensitivity in controlling expansion and differentiation of CD8+ T cells with different TCR avidities toward peptide/MHC ligands, during the response to pathogens. In both polyclonal and monoclonal settings, through use of mice with selective loss of IFN- γ receptor on CD8+ T cells, the authors find that IFN- γ sensitivity promotes the generation of CD8+ T cells with lower avidities during the effector phase of the response, while favoring establishment of higher avidity cells in the memory pool (capable of recall responses). This involves CD8+ T cells sensing IFN- γ produced early in the infection, and the authors provide evidence that the source of this is virtual memory CD8+ T cells (TVM). Together, these data indicate that IFN- γ - aside from its well-known direct role in control of pathogens - regulates how CD8+ T cells with low vs high TCR avidity will contribute to the effector vs memory pools. In addition, the authors show that this regulation has material consequences (as indicated by mice lacking IFN- γ receptor on CD8+ T cells displaying enhanced resistance to primary influenza infection) and provide evidence that there may be analogous contributions of TVM-like cells in humans.

The data are, for the most part, compelling and high quality, and the findings are interesting. This builds on previous data that indicated IFN- γ sensitivity alters the maintenance or trafficking of CD8+ T cells, but the authors findings are distinct and suggest a role in fine-tuning the contribution of CD8+ T cells of distinct avidities into short- vs long-lived populations.

The most clear-cut and convincing data are those showing that IFN- γ receptor expression has distinct impacts on the response of OT-I T cells to ligands of different affinities, at different stages of the responses: IFN- γ receptor deficiency favoring the OT-I response to high but not low affinity ligands at the effector stage of the primary response, with the opposite pattern for recall responses (Figs 5H, 6A-C). This is reasonably well echoed in analysis of polyclonal responses. The findings on the basis for the initial production of IFN- γ were less compelling, however and led to concerns that should be addressed.

1) The authors propose that TVM are the relevant source of the "early" IFN- γ impacting antigen-specific CD8+ T cells. Prior data indicate that memory-phenotype CD8+ T cells are capable of innate-like functions, including production of IFN- γ in response to inflammatory cytokines. However, the authors propose instead that this response is antigen-specific, involving TCR stimulation of TVM. Their most direct approach to test this idea used adoptive transfer of OT-I T cells into mice, which were then challenged with LM bearing the OT-I cognate ligand (OVA), or a control antigen (gp33) (Fig. 4I). However, the authors do not show any data on the frequencies or numbers of OT-I TVM that were engrafted by this protocol, nor how this population was unambiguously identified 24 hours after the two different infections. What markers were used to identify the "OT-I TVM" pool in adoptive recipients? Did the frequency of that population differ in the mice challenged with LM-OVA vs LM-gp33? If there is an apparent increase in the frequency of "TVM" in the LM-OVA (but not LM-gp33) infected mice, there is a substantial risk that this includes OT-I cells which responded to OVA as naïve OT-I cells and have rapidly acquired activated/memory phenotypic characteristics. More data and information about these experiments are needed.

2) It is striking that the authors find no apparent role for NK cells - the major early producers of IFN- γ (Fig. S3) - in driving the avidity skewing of antigen-responsive CD8+ T cells (Fig. 3F). Do the authors conclude that this reflects NK and CD8+ T cells being in distinct anatomic locations? Another interpretation is that, rather than a response to paracrine IFN- γ , at least some CD8+ T cells may selectively respond to autocrine IFN- γ . This seems eminently testable in the mixed chimeras described in Fig. 3: The control chimeras involved donors that were a mixture of IFN- γ deficient and WT donors. While not stated, presumably these were congenically distinct donors (otherwise there would be no way of checking that the chimeras were appropriately balanced), in which case the authors can distinguish whether there is a difference in the apparent TCR avidity of cells that can/cannot make a source of autocrine IFN- γ . If the authors find that IFN- γ -deficient (like IFN- γ receptor-deficient) CD8+ T cells showed enhanced TCR avidity - despite the presence of WT CD8+ T cells in the same chimera - this would indicate a key role for autocrine IFN- γ responsiveness, significantly enhancing the impact of the report.

3) It would be valuable for the authors to return to the striking findings in Fig. 1, which showed markedly enhanced resistance to influenza in mice where CD8+ T cells lack IFN- γ receptor.

Presumably, the authors would predict that the vulnerability of WT mice to high dose X31-OVA (Fig. 1C,D) might be "rescued" by adoptive transfer of IFN- γ -deficient but not WT OT-I cells, since the former would be favored for effector-phase expansion. Is this in fact the case?

Uhl et al., RESPONSE TO REVIEWER COMMENTS

Reviewer #1 (expert in CD8+ T cell memory):

In this study, the authors demonstrated that IFN-g sensing by CD8+ T cells regulates avidity of the CD8+ T cell population during primary and memory responses. The authors showed that IFN-g limits the expansion of CD8+ T cells with high avidity for their cognate peptide, while the expansion of CD8+ T cells with low avidity is preserved. The authors also demonstrated that virtual memory T (Tvm) cells are major cellular sources of IFN-g and that IFN-g works in a paracrine manner. Finally, they showed that IFN-g sensing by CD8+ T cells increases the avidity during the memory response. In this study, the authors tried to reveal the direct influence of IFN-g signaling on CD8+ T cells encountered with cognate antigens, especially focusing on how varying CD8+ T cells with a breadth of avidities are selected and expanded throughout primary and memory immune responses. Although they investigated this subject by using various mouse models including IFN-g receptor KO mouse (CD8-IFN-gRKO), further experiments and explanations are required to improve this study.

Thank you for the thorough assessment of our manuscript. We believe the answers and further experiments provided below greatly improved the manuscript and hope it mitigates the queries of Reviewer 1.

1. The authors showed that IFN-g sensing by CD8+ T cells decreases avidity during primary responses. One of possibilities is that IFN-g selectively eliminates CD8+ T cells with high avidity by inducing apoptosis or cell death, thereby decreasing avidity of the antigen-specific CD8+ T cell population. Please test this hypothesis by performing further experiments.

We tested this hypothesis both in vivo and in vitro by analysing apoptosis in OTI cells primed by their high avidity peptide N4. In vitro, WT OTI cells were primed with N4 for 36 hrs in the presence of increasing doses of IFN γ . In this system, IFN γ did not have a direct effect on OTI apoptosis, quantified by Annexin V staining. In vivo, OTI T cells were transferred in WT mice, which were infected with Listeria-N4. After 24 hrs, mice were treated with anti-IFN γ . At the peak of the response, apoptosis of OTI cells was analysed by Annexin V staining. Here again, we did not detect any difference in apoptosis. Overall, this new data (Fig.5I and J) strongly suggests that IFN γ does not lead to apoptosis of high affinity T cells, both during priming and at the peak of the response.

Our sequencing data suggest that IFN γ rather affected the proliferation of effector cells. In an effort to provide a molecular mechanism to explain the difference in avidity observed between WT and CD8-IFN γ RKO mice, we analysed whether proliferation was affected by IFN γ when cells were primed with high- or low-avidity peptides. To do so, WT and IFN γ RKO OTI were ad-mixed and transferred in WT recipient mice, which were infected with LM-N4, -Q4H7 or -V4. EdU was injected the day before harvest. As expected from the sequencing data (Fig.5G), at the peak of the effector response, IFN γ sensing by OTI cells inhibited proliferation when cells were primed with high-, but not low-avidity (Fig.5K).

2. In Figure 1, please show quantitative kinetics of OVA-specific CD8+ T cells in WT and KO mice and kinetics of viral load.

As suggested, we are now providing quantitative kinetics of OVA-specific CD8 T cells in WT and KO mice (Fig.1I), which confirms that there are no obvious differences in overall CD8 T cell recruitment in the lung at the peak of the response. In addition, we also quantified the viral load in the lung of WT and KO mice and found an accelerated viral clearance in KO mice compared to WT mice (Fig.1D).

3. In Figure 1G, please examine granzyme B and perforin in addition to LAMP-1.

We now provide quantification of Granzyme B and perforin expression, as suggested by reviewer 1. We do not detect any difference in their expression. This new data is presented in Fig.1G and H.

4. In the evaluation of avidity by MHC-I tetramer binding, please explain a justification of normalization by the CD3 expression.

Thank you for pointing out the lack of explanation for the normalization method used to extract the relative avidity. We clarified this as below:

“Differences in tetramer staining can simply reflect differences in TCR expression or TCR down-regulation following stimulation. To ensure that differences in tetramer binding between WT and CD8-IFN γ RKO mice were not the result of differential TCR expression, we normalized the MFI of tetramer with the one of CD3. This has been used previously in other studies and is named relative avidity.”

5. In NK cell depletion analysis, please measure EC50 for more accurate analysis of avidity.

As suggested by Reviewer 1, we have now performed EC50 measures of CD8 T-cells following NK depletion. We did not observe differences in IFN γ production and EC50 (Fig.3F and G), which is consistent with the original data and relative avidity measures (Fig.S3D). We therefore concluded that NK cells (and therefore IFN γ -derived NK cells) do not regulate T cell avidity.

6. In Figure 4, the UMAP projection split by control mice and infected mice is recommended to be shown.

The UMAP is now split between control and LM-OVA infected mice (Fig.4A).

7. In Figure 4B, the scale axis of the color bar may be opposite (as naïve cells are known to have a high expression of *Ccr7*, *Sell* and *Lef1*).

This was indeed a mistake. Thank you for pointing this out. The color bar has been corrected.

8. In Figure 4D, as the 10x genomics cell ranger output files are a sparse matrix, obvious criteria for determining IFN-g+ cells needs to be shown (for example, in violin plot or feature scatter plots).

We added a violin plot that shows IFN γ expression in naïve vs LM-OVA (new Fig.4D). We also clarified in the text that we selected all cells that displayed IFN γ transcripts and analysed the proportion of cell types/clusters within. This data is consistent with flow cytometry data. As such, we are confident that TVM are the main IFN γ producers during priming.

9. In Figure 4M, it will be better to show if *Tvm* cells in this public dataset express *Ifng*.

In this dataset, we did not reliably detect IFN γ mRNA (not only by TVM but by all CD8 T cells), suggesting that cells in the blood were not actively producing IFN γ (Fig.A). The little IFN γ expression we could detect was driven by 3 samples. We therefore could not perform the analysis suggested.

10. In Figure 5B, the CX3CR1⁺ MPEC still show some expression of effector-related markers like *Klrg1* and *Tbx21*, so the authors should examine additional markers to confirm these population as MPEC population. Additional public dataset may be used for further validation.

The reviewer is right, the CX3CR1⁺ MPEC subset contains both effector and memory markers. Importantly, this cluster is stable and present at higher resolution, suggesting that those are the features of the cells within this cluster. As suggested, we confirmed that this cluster exhibits a memory signature, together with an intermediate scoring of effector signature, signatures both extracted from publicly available datasets (Fig.S5D-E). We believe this is not an intermediate subset based on the findings of the Gerlach group (Immunity 2016, 2023) demonstrating that intermediate expression of CX3CR1 characterises a distinct T cell differentiation state, and we named it MPEC based on the fact that those cells preferentially become long-term memory cells surveilling peripheral tissues (Tpm). The CX3CR1^{high} population is the only one giving rise to cytotoxic cells (Zwijnenburg et al, Immunity 2023)

11. In Figure 5C, CD8-IFN-gRKO mice still seems to have *Ifngr1* expression. Authors need to show the actual expression level of *Ifngr1* in CD8-IFN-gRKO mice.

We now provide example histograms showing that CD8 T cells from CD8-IFN-gRKO mice display similar IFN γ R1 profile than IFN γ RKO mice (Fig.S1C). We believe the residual IFN γ R1 signal observed in our transcriptomics dataset might be due to the fact that the floxed mice possess loxP sites flanking exon 3 and 4 of the *Ifngr1* gene, allowing transcription of the first exons.

12. Figure legend for Fig. 1 mentions that “Data are pooled from 3 (A-D, G, J, n = 11-13) or 4 (F, H-I) independent experiments. How many mice were used per experiment? What is the n number for Fig. 1F, H-I? n number needs to be more clearly indicated.

We apologise for the lack of clarity, we are now providing the exact number of mice and independent experiments throughout the manuscript.

13. Several studies have previously demonstrated that KO of IFNGR1 does in fact affect CD8⁺ T cell proliferation, differentiation, and effector functions (Science 2000, 290(5495):1354-1357; J Exp Med 2005, 201(7):1053-1059; J Immunol 2010, 184(6):2855-2862; J Immunol 2012, 189(2):659-668), though the exact positive or negative impact of IFN-g on CD8⁺ T cells remains to be conclusively determined. Yet, no such impact on cell expansion or effector functions were reported by the authors as seen in Figure 1. The authors should provide some explanatory discussion of this discrepancy between the findings of this and study others in the manuscript.

The reviewer is right, it is important to put our data back into context. There are multiple hypotheses to explain the discrepancies between our manuscript and other publications. Some studies use a whole different system, either in vitro stimulation, a different trigger, antigen system, or full KO mice. The use of a full KO will inhibit IFN γ -sensing of all cells, as all cells express the IFN γ R, which might indirectly impact T cell differentiation or expansion. In addition, many studies use OTI and OVA (N4 system), which skews that system and does not take into account the differential effect of IFN γ on low-affinity T cells. Our current manuscript clearly shows IFN γ has a distinct effect both in terms of expansion and differentiation according to the TCR affinity at which cells were primed. We do describe an effect of IFN γ on OTI expansion, proliferation and differentiation in the manuscript (Fig.5G,H,K, Fig.6A-D), but the endogenous data, being the sum of all high- and low-affinity T cells, will integrate the sometimes opposite effects of IFN γ on high- and low-affinity T cells, cancelling each other out at the population level. We believe this is an important point of our manuscript, showing that conclusions drawn exclusively using transgenic TCR might not reflect the endogenous response (or reflect just one aspect of the endogenous response). This is now further discussed in the discussion section as such:

“Several studies have previously demonstrated that contradictory role of IFN- γ on CD8⁺ T cell proliferation, differentiation, and effector functions^{23, 27, 28, 30, 53, 54}. Some might be explained by the use of full IFN- γ RKO mice, where IFN- γ might target other cells that could indirectly affect T-cell function,

or a different immunization protocol. But in addition, our data provide some further explanation as for why those discrepancies exist. Indeed, we provide evidence that IFN- γ have a distinct effect on high-versus low-avidity T-cells. As such, studies that used TCR transgenic mice might have revealed only a part of IFN- γ function. This is evident when we compared the endogenous response versus the TCR transgenic response for example (Fig.5D-E versus Fig.6A-B).”

Reviewer #2 (expert in T cell-mediated immune responses):

This article uses a combination of mice genetically lacking expression of IFN γ R selectively on CD8 T cells, as well as a series of BM chimeras to dissect the impact of IFN γ sensing by CD8 T cells on their avidity. Using the CD8-IFN γ R KO mice they show that a loss of IFN γ sensing by CD8 T cells has relatively little effect on response magnitude but results in an increased TCR affinity (by comparing tetramer staining while controlling for TCR levels) following infection with either influenza or Listeria monocytogenes. This also translated to a shift in functional avidity. This was also supported by a nice experiment (Fig 5H), where Lm-OVAp variants were used to show a selective impact of IFN γ RKO on high avidity compared to low avidity cells in the effector response.

Using neutralizing Ab experiments the authors showed that early (16-24h) administration of anti-IFN γ Abs also increased the avidity of T cells in the effector response, while later (d5/6) did not, suggesting that IFN γ sensing during priming was important. While this seems reasonable, a caveat here is that neutralizing Ab can persist for up to a week so the critical timing here is unclear.

The reviewer is right, antibodies can persist for multiple days. However, we now provide evidence that, during the primary response, 2 waves of IFN γ expression exist, one at 24h and one picking up at day 6 post infection. As such, even if the antibody injected at 24hrs was still present during the second wave, the fact that no differences in avidity were observed when we injected antibodies at day 5/6 eliminates the possibility that IFN γ is important at a later stage for regulating avidity.

Through NK depletion and mixed BM chimeras the study found that CD8 production of IFN γ was critical for this effect. The authors showed that the main producers of IFN γ after Lm infection were Tvm cells. Importantly, while the phenotype of the cells (CD44^{hi}, CD49d^{lo}, CD122^{hi}) is clearly a Tvm phenotype in unchallenged mice, can the authors be certain these are not early activated cells at 24h after Lm-OVA infection? Do they find a similar frequency of these cells in the control mice?

We now provide quantification that similar frequencies of T_{VM} are found 24hrs post-infection, suggesting that the phenotype of those cells is still stable at this time point (Fig.S4C (endogenous) and 4G (OT-I)).

They showed that Tvm cells produce the largest amounts of IFN γ when stimulated through their TCR in addition to cytokines, however I note that they do not include a condition of combined IL-12 and IL-18 stimulation (standard for stimulating Tvm production of IFN γ) in the absence of TCR stimulation.

This was a mistake as the figure should have included this condition and we apologise for this. We are now providing the complete figure showing that TCR+IL12+IL18 elicited greater IFN γ production compared to IL12+IL18 only (Fig.4H).

The authors then conclude that Tvm cells need to be stimulated through their TCR (rather than in a bystander manner) in order to produce the IFN γ that regulates avidity. But they adoptively transfer total OTI cells from OTI x GREAT mice and show that those cells produce IFN γ upon infection with Lm-OVA. How are they attributing this to Tvm cells? The results in Fig 4I and J simply seem to indicate that they have an antigen-specific response by OTI cells to Lm-OVA. Of course, once stimulated by OVA, the cells would not exhibit a Tvm phenotype anymore as they would be antigen-experienced.

We are now providing evidence that the OTI cells producing IFN γ are mainly T_{VM} (Fig.4K). We agree with Reviewer 2 that it indicates an antigen-specific response, but this does not invalidate the fact that, although TCR priming is required for maximum IFN γ by T_{VM}, it has to be indeed antigen-specific.

Because OTI (including T_{VM} , which are about 20% of OTI cells in a naïve mouse (Fig.S4C)), do not produce IFN γ following LM-gp33 infection (Fig.S4J), we concluded that bystander activation was not enough to fully prime T_{VM} to produce IFN γ . To formally prove that T_{VM} have a higher capacity to produce IFN γ than naïve T cells, we made use of the OTI system again. We sorted naïve (CD44^{neg}) and T_{VM} (CD44⁺CD49^{dneg}) OTI cells and transferred them in WT recipients. Mice were then infected with LM-OVA. A higher percentage of T_{VM} cells were producing IFN γ compared to naïve T cells (Fig.4L). Most importantly, while we agree with the reviewer that T_{VM} will eventually become CD49^{d+} because they have been primed by their TCR, the phenotype of transferred cells is still stable 24hrs post infection, with naïve T cells mainly retaining their naïve phenotype and T_{VM} mainly retaining their T_{VM} phenotype at this time point (Fig.4M). This is consistent with what is observed for the endogenous response, and confirms that T_{VM} are the main source of T-cell-induced IFN γ during the early wave of IFN γ production.

There are many other issues with this study. Generally, the study is underpowered and conclusions have been drawn on relatively limited datasets or on experiments that provide supportive but not definitive results. Details of some other key issues are as follows:

We believe that our study was adequately powered. There might have been some lack of clarity, where often dots represented experiments and not mice. To avoid any confusion, we are not presenting data as number of mice instead of number of experiments. Further details are provided below.

Were the control mice (WT) used in this study IFN- γ R1 flox/flox mice? Given the relatively mild phenotype differences after x31 infection, it is important to demonstrate that this isn't a consequence of minor differences in background.

We agree that this is an important point. We are always using littermates, either CD8-Cre or IFN γ Rf/f mice. We apologise if this point was not clear. We clarified this in the methods section and replaced "WT" by "Ctrl" in the figures to avoid any confusion.

In addition, in my extensive experience, HKx31 strain of flu is only moderately virulent and is never lethal in B6 mice. Is Fig 1D actually showing death or is it showing a humane endpoint and if the latter please clearly articulate what this is in the legend and in the methods. It is important that this distinction is also made in the text. In addition, the legend indicates that mice were infected with 10e5 pfu while the methods indicate that mice were infected with 4 x 10e4 or 4 x 10e5 pfu. Please clarify.

This is an interesting point. When we used X31 in the US, we had a similar experience as reported by the reviewer and did not observe death. However, X31 can kill mice in our current facility in the UK. We have many Condition 18 (a form in the UK to report mice that were found dead instead of killed before they reached humane endpoint) to illustrate this. It is unclear why this is the case. Microbiota composition and the relative presence of pathogens in a given facility are often suggested as a cause for those discrepancies. Regardless, we agree with the reviewer that we need to clarify that most mice were not left to die but killed according to humane end point (although we would like to point out that in our experience, mice would not recover if they displayed signs of ill-health, lack of grooming, no eating or drinking or lack of movement, which were our human timepoint). We also clarified the X31 dose used throughout.

For all figures, the number of times each of these experiments has been performed needs to be included in the figure legends. For example, Fig. 1F, H, and I have only 4 mice per group, and should be repeated to firm up conclusions.

We are now displaying the number of mice and increased the number of mice where necessary.

Fig S2 A-C, why does panel A have so few data points when these data are derived from the same experiments as those shown in panels B and C? All data points should be included in A.

We now equalized the number of mice, ensuring they are coming from the same experiments.

Similarly, for Fig 3A-C. Why does panel C have so few datapoints when this data comes from the same experiment as those depicted in A and B?

We apologize for the lack of clarity. We are now providing more samples for Fig.3C, but in a few samples, the CD3 staining did not work sufficiently well to confidently quantify the MFI and in those conditions, we used CD8, which also correlates with TCR expression. We are now making this clear in the text : “ Differences in tetramer staining can simply reflect differences in TCR expression or TCR down-regulation following stimulation. To ensure that differences in tetramer binding between WT and CD8-IFN γ RKO mice would not be the result of differential TCR expression, we normalized the MFI of tetramer with the one of CD3 or CD8. This has been used previously in other studies and is named relative avidity, a measure of TCR affinity and/or avidity at the population level ⁵.”

Also, for Fig 2E-J n= 2 or 3. Firstly, in the legend it is stated “each dot represents an experiment”. What is meant by this? Why doesn’t each dot represent the data from one mouse? Secondly, it is impossible to get a statistically significant value when n<3 so I do not understand how this was obtained for Figs 2F and 2H. Additional datapoints must be included here.

We wanted to display experiments rather than mice, to avoid any batch effect from one experiment. Using this display, we detected an increased avidity for every experiment we performed. We thought it was important, as we were assessing the endogenous response, which can be highly variable. As suggested by Reviewer 2, we are now displaying data as mice rather than experiments, which shows the same result.

For Fig. S2A, E and F there are too few data points to definitively state difference or no difference, especially if they have been performed only once. If performed more than once, an example of an additional experiment should be included in supplementary data to give confidence that this finding (with a current n of only 3 or 4) is reproducible.

We have now increased the number of mice to improve the confidence in our data.

The authors state on top of page 5 “as the increased tetramer binding in CD8-IFN- γ RKO CD8+ T-cells was already present 5 days after infection (Fig.S2G).” This is a dubious call. The datasets look to be of almost identical magnitudes with the exception of one higher value in the KO mice, and this difference is almost certainly not significant. Revise this interpretation.

The reviewer is right. We have now revised this interpretation as such, which we believe is more accurate:

“Increased tetramer binding in CD8-IFN- γ RKO CD8+ T-cells was not present before 5 days after infection (Fig.S2H), suggesting that IFN- γ did not inhibit priming of high-affinity T-cells.”

Fig 2. A-D, what is the gating strategy here? What about lymphocyte and live gating? It should be included in supp data, including representative splenic tetramer staining.

We used the exact same strategy throughout the manuscript, which was depicted in Fig.S1.

What does it mean that mice were treated with anti-IFN γ Ab at days 5-6? Is this one or two administrations? This should be included in methods.

Thank you for pointing this out. We clarified that we treated the mice with anti-IFN γ twice, one day apart on days 5 and 6 in the methods section.

The cited paper (28: Tewari et al, 2007) for the two waves of IFN γ production during infection seems incorrect. Haring et al, 2005, J Immunol suggests that CD8 T cells are responsive to IFN γ within 12 h after Lm infection. The authors should cite the correct paper or alter their statement on IFN γ production after Lm infection.

Thank you for picking this up. We added the correct citations, but we also included our own quantification in Fig.S3A, which is important to clarify the reason for blocking IFN γ at day 1 and day

5+6. We confirmed that we indeed can detect 2 waves of IFN γ production following LM infection in situ. That is not consistent with the Haring et al study, which does not detect the second wave, but the second wave has been described in multiple papers (see for example Buchmeier and Schreiber, PNAS 1985, Krummel et al., PNAS 2018)

In Haring et al, to investigate responsiveness, they restimulate cells in vitro. This comes with the caveat that if cells are receiving a high dose of IFN γ in vivo, they might become unresponsive to further restimulation in vitro, which does not mean that they were not responsive at this time point in vivo, but rather that the receptor is saturated. In our manuscript, we clearly demonstrate that CD8 T-cells can phosphorylate Stat1 24hrs post infection, and that this is, at least partly, driven by IFN γ (Fig.3H of original manuscript).

For Figure 2A, F and H, and Fig 3H these data points should not be joined by lines. These are not paired analyses and there is nothing connecting individual WT mice with individual KO mice. These data should be shown independently.

We joined lines because dots represented experiments rather than mice. We joined conditions that were from the same experiments. In this case, we believe they were therefore paired. However, to be consistent, we are now providing data as number of mice rather than experiments.

With regard to Fig 3H, the IFN γ signalling data should not be 'normalized' to the IFN-g KO. This can be used as a background that can be subtracted but fold difference from (theoretical) zero is unachievable. For this reason, the statement at the top of page 6: "Ablation of CD8-induced IFN- γ led to 40% inhibition of IFN- γ signaling (Fig.3H)" may be inaccurate and should be recalculated.

As suggested, we are now providing data where we subtracted the background (from IFN γ KO samples) rather than normalizing by fold induction (*new Fig.3I*). We still observe a reduction of in pStat1 in the same order of magnitude.

Similarly, the lines in Fig 6A and B should be removed. These are not continuous samples.

As advised by reviewer 2, the lines have been removed.

The analysis of human cells shown in Fig 4 K-M concludes that elevated frequencies of T_{VM} cells in humans corresponds to severe COVID. Firstly, how many patient samples is this? The legend indicates for M that each dot is a patient but there are no dots here. Secondly, this could equally indicate that severe disease, characterised by heightened inflammation, drives elevated T_{VM} numbers, rather than the other way around.

We apologise for not displaying the number of patients, this now appears in the legend (n=60). Because many samples had no detectable T_{VM}, we are also adding a new graph where we focussed on samples that contain T_{VM} to increase confidence (COVID mild = 12, COVID severe = 25). The data now appears in Fig.S4M-P.

We agree with reviewer 2 that T_{VM} proportion only correlate with disease severity, and we have no way to know with this data what the initial trigger is. We believe we did not claim that there was an actual causation ("Analysis of the relationship between T_{VM} and disease severity revealed that enhanced frequency of T_{VM} is associated with severe COVID disease"). However, to make this point clearer, we added the sentence: "Because inflammation induced during severe disease could drive elevated T_{VM} numbers, it also correlates with our finding that CD8⁺ T-cells mainly sense IFN- γ produced by T_{VM}, leading to decreased avidity of the primary response and a subsequently curtailed response during infection."

For Fig 6C, why were the mice challenged with Lm-OVA? If the point of this experiment was to determine the extent to which priming with different avidities influenced memory formation, why not just look in the memory pool at d60 after Lm-OVA priming? This would eliminate the confounding influence of differential expansion upon rechallenge.

We initially did not include this data, as the number of OT-I was not detectable before recall, especially for low-avidity peptides. To answer this question, we analyzed the proportion of OT-I at later time (day 50+ after infection) when we transferred a higher number of OTI (500,000). Before re-challenge, the ratio between WT and IFN- γ RKO OT-I T-cell was similar to that which was observed at the peak of the response for high-avidity priming (Fig.6C). For OT-I cells primed with lower-avidity, however, the number of WT and IFN- γ RKO OT-I T-cells equalized, suggesting that IFN- γ RKO OT-I T-cells might preferentially differentiate towards memory (Fig.6C). This was further accentuated after recall, as expansion of IFN- γ RKO OT-I T-cells primed with low-affinity peptides was increased compared to WT during recall responses (Fig.6D). Priming with high-affinity peptides, however, led to a slightly decreased frequency of IFN- γ RKO over WT OT-I T-cells during recall responses (Fig.6D).

Minor points

Page 2, last paragraph - "...regulates clonal selection to coordinate avidity....", and "...skewing their differentiation towards effector cells"

This sentence was edited: Here, we present evidence that direct IFN- γ sensing by CD8⁺ T-cells regulates clonal selection **by coordinating** avidity and differentiation during an immune response

The statement in paragraph 4 of page 5 "This demonstrated that the increased avidity observed in CD8-IFN- γ RKO mice was not the result of indirect increased IFN- γ signalling in other cells." is not technically correct. The earlier experiments where IFN γ R was knocked out only on CD8 T cells made this point.

We agree with the reviewer that IFN γ R deletion in CD8 T cells was the correct strategy to show that the decreased avidity was due to IFN γ signalling in CD8 T cells. However, there was still the possibility that IFN γ R deletion in T cells would induce an increase in IFN γ concentration and IFN γ bioavailability in the spleen, given the high proportion of CD8 T cells in the spleen that would not consume IFN γ . As such, there was a slight risk that the phenotype we observed was confounded by a potential increase in IFN γ availability, resulting in enhanced signalling in other cell types. Inhibiting IFN γ allowed us to exclude this and to univocally demonstrate that IFN γ sensing by CD8 T cells decreases the average avidity of their response. We added this sentence to improve clarity:

"Indeed, because the majority of cells in the spleen are T-cells, there was a risk that IFN- γ R deletion in T-cells resulted in increased IFN- γ bio-availability and thereby increasing IFN- γ signaling in neighboring cells, indirectly increasing T-cell avidity."

This manuscript should cite the recent paper by Dirk Busch's group - Straub et al (Immunity) 2023 - which clearly demonstrates the proliferative advantage of high avidity T cells, their relative imperviousness to the multitude of low avidity responses, and in particular, the proliferative and protective capacity of low avidity T cells recruited into primary memory, upon rechallenge with a mutant peptide for which they exhibit higher avidity than for the original epitope. In the Discussion, the authors should speculate what the implications of this study are for their current data set.

We agree with Reviewer 2. This is an outstanding study that got published after we sent our manuscript for consideration in Nat. Comms. We believe our data are not in contradiction with Dirk Busch's paper, but complement it. We discussed it as followed:

"Low-affinity and avidity T-cells are intrinsically skewed to become memory T-cells where they were suggested to be important for memory responses towards mutated pathogens. This has been recently confirmed by the group of D. Busch in a study that demonstrated that low-avidity T-cells get recruited and expand as long as they pass a clear TCR avidity threshold. This results in the recruitment of a broad polyclonal repertoire where low-avidity T-cells allow for flexible secondary responses to mutant epitopes. In this context, IFN- γ regulates the other end of the spectrum by restricting the expansion of high-avidity T-cells, which, as a result, ensures that the avidity of the T-cell response is conserved and high enough during memory stages to enable efficient long-term responses towards native but also

mutated pathogen, where T-cells that were initially recruited as low-avidity can become major responders.”

Reviewer #3 (expert in homeostasis, trafficking, and differentiation of CD8+ T cells):

The authors explore the role of IFN- γ sensitivity in controlling expansion and differentiation of CD8+ T cells with different TCR avidities toward peptide/MHC ligands, during the response to pathogens. In both polyclonal and monoclonal settings, through use of mice with selective loss of IFN- γ receptor on CD8+ T cells, the authors find that IFN- γ sensitivity promotes the generation of CD8+ T cells with lower avidities during the effector phase of the response, while favoring establishment of higher avidity cells in the memory pool (capable of recall responses). This involves CD8+ T cells sensing IFN- γ produced early in the infection, and the authors provide evidence that the source of this is virtual memory CD8+ T cells (TVM). Together, these data indicate that IFN- γ – aside from its well-known direct role in control of pathogens – regulates how CD8+ T cells with low vs high TCR avidity will contribute to the effector vs memory pools. In addition, the authors show that this regulation has material consequences (as indicated by mice lacking IFN- γ receptor on CD8+ T cells displaying enhanced resistance to primary influenza infection) and provide evidence that there may be analogous contributions of TVM-like cells in humans.

The data are, for the most part, compelling and high quality, and the findings are interesting. This builds on previous data that indicated IFN- γ sensitivity alters the maintenance or trafficking of CD8+ T cells, but the authors findings are distinct and suggest a role in fine-tuning the contribution of CD8+ T cells of distinct avidities into short- vs long-lived populations.

The most clear-cut and convincing data are those showing that IFN- γ receptor expression has distinct impacts on the response of OT-I T cells to ligands of different affinities, at different stages of the responses: IFN- γ receptor deficiency favoring the OT-I response to high but not low affinity ligands at the effector stage of the primary response, with the opposite pattern for recall responses (Figs 5H, 6A-C). This is reasonably well echoed in analysis of polyclonal responses. The findings on the basis for the initial production of IFN- γ were less compelling, however and led to concerns that should be addressed.

We would like to thank the reviewer for the positive evaluation of our manuscript and for emphasising the novel parts of our work. Please find the answer of the reviewer’s concerns below

1) The authors propose that TVM are the relevant source of the “early” IFN- γ impacting antigen-specific CD8+ T cells. Prior data indicate that memory-phenotype CD8+ T cells are capable of innate-like functions, including production of IFN- γ in response to inflammatory cytokines. However, the authors propose instead that this response is antigen-specific, involving TCR stimulation of TVM. Their most direct approach to test this idea used adoptive transfer of OT-I T cells into mice, which were then challenged with LM bearing the OT-I cognate ligand (OVA), or a control antigen (gp33) (Fig. 4I). However, the authors do not show any data on the frequencies or numbers of OT-I TVM that were engrafted by this protocol, nor how this population was unambiguously identified 24 hours after the two different infections. What markers were used to identify the “OT-I TVM” pool in adoptive recipients? Did the frequency of that population differ in the mice challenged with LM-OVA vs LM-gp33? If there is an apparent increase in the frequency of “TVM” in the LM-OVA (but not LM-gp33) infected mice, there is a substantial risk that this includes OT-I cells which responded to OVA as naïve OT-I cells and have rapidly acquired activated/memory phenotypic characteristics. More data and information about these experiments are needed.

This is an important point that we decided to address from multiple angles. First, as suggested by the reviewer, we confirmed that IFN γ production by OTI cells 24hrs following LM-OVA is also driven by T_{VM} (Fig.4K). We also confirmed that the proportion of naïve, T_{VM} and TM does not change 24 hrs following infection, suggesting that the T_{VM} phenotype following infection is not due to naïve cells up-regulating CD44 and down-regulating CD49d (Fig S4C and S4J). In a second set of experiments, we directly tested the fact that T_{VM} have a higher capacity to produce IFN γ 24hrs after priming compared to naïve T cells.

To do so, we isolated and sorted naïve and T_{VM} OT-I T cells, which were then adoptively transferred in WT mice. Mice were then infected with LM-OVA, and IFN γ production and phenotype of those cells were analysed after 24hrs. We could confirm that a higher proportion of T_{VM} cells produced IFN γ compared to naïve T cells (Fig.4L). In addition, we also confirmed that at this time point, T_{VM} retain the same phenotype (CD44 $^{+}$ CD49d $^{-}$), whereas naïve T cells become CD44 $^{+}$ CD49d $^{+}$ (Fig.4M). Overall, we believe this new data strengthen our initial conclusion that T_{VM} are the main producers of early IFN γ production.

2) It is striking that the authors find no apparent role for NK cells – the major early producers of IFN- γ (Fig. S3) – in driving the avidity skewing of antigen-responsive CD8 $^{+}$ T cells (Fig. 3F). Do the authors conclude that this reflects NK and CD8 $^{+}$ T cells being in distinct anatomic locations? Another interpretation is that, rather than a response to paracrine IFN- γ , at least some CD8 $^{+}$ T cells may selectively respond to autocrine IFN- γ . This seems eminently testable in the mixed chimeras described in Fig. 3: The control chimeras involved donors that were a mixture of IFN- γ deficient and WT donors. While not stated, presumably these were congenically distinct donors (otherwise there would be no way of checking that the chimeras were appropriately balanced), in which case the authors can distinguish whether there is a difference in the apparent TCR avidity of cells that can/cannot make a source of autocrine IFN- γ . If the authors find that IFN- γ -deficient (like IFN- γ receptor-deficient) CD8 $^{+}$ T cells showed enhanced TCR avidity – despite the presence of WT CD8 $^{+}$ T cells in the same chimera – this would indicate a key role for autocrine IFN- γ responsiveness, significantly enhancing the impact of the report.

We agree that Reviewer 3 that the fact that NK cells were not involved in T cell avidity was surprising. This was even more surprising as IFN γ -producing CD8 T cells are in fact found in the same microenvironment as NK cells, which get into the white pulp after listeria infection (Fig.B).

Figure B- NK and IFN γ + CD8 T cells are present in the same microenvironment. WT mice were transferred with CellTracker Orange (CMTMR)-labelled OTI cells (red), infected with LMOVA, and killed 24 h after infection. Mice were treated with brefeldin A 6 h before harvest. Frozen spleen sections were stained for NK (green) and IFN γ (blue). Photographs are representative examples of IFN- γ -producing NK and OTI cells.

While the reviewer's idea to reanalyse our chimera data to address the question of autocrine versus paracrine IFN γ signalling was ideal, we unfortunately used separate stains (for the same mouse) to investigate avidity and chimerism. We as such couldn't perform that analysis. However, we used our scRNAseq data to get that information. We used NicheNet to analyze cell-cell communication between the different CD8 T cell subsets present 24hrs post-infection. First of all, we found that IFN γ was the top channel used by T cells to communicate (Fig.S4C,E), confirming the importance of our study. Then, if IFN γ signalling was autocrine, we would expect that T_{VM} would primarily display enhanced IFN γ signalling compared to other subsets. NicheNet analysis suggested that, while T_{VM} were the main "sender" (Fig.S4D,F), all CD8 T cells could receive IFN γ and signal (Fig.S4C,E). In addition, computing an IFN γ signalling score revealed that T_{VM} had a lower score than other T cells, suggesting they were not the main subset sensing IFN γ (Fig.S4G), which we hypothesize was due to the fact that they express less of IFN γ R (both IFNGR1 and IFNGR2, Fig.S4H). Overall, our data is consistent with paracrine rather

than autocrine IFN γ signalling, which is in agreement with already published data (PMCID: PMC6233119). We, however, still believe this is an important study. Our study is one of the first providing mechanistic evidence that the average avidity of the T cell response is actively regulated (by IFN γ) and important for the optimal tread-off between initial viral control and memory formation.

3) It would be valuable for the authors to return to the striking findings in Fig. 1, which showed markedly enhanced resistance to influenza in mice where CD8+ T cells lack IFN- γ receptor. Presumably, the authors would predict that the vulnerability of WT mice to high dose X31-OVA (Fig. 1C,D) might be "rescued" by adoptive transfer of IFN- γ -deficient but not WT OT-I cells, since the former would be favored for effector-phase expansion. Is this in fact the case?

We agree with Reviewer 3 that inhibiting sensing of IFN γ by OTI cells should enhance their potential to rescue/ameliorate the influenza response of WT mice, given our findings that IFN γ sensing by OTI cells restricts their expansion (Fig.5H) and proliferation (Fig.5K) when primed with high-affinity peptide. To test this, we transferred WT or IFN γ RKO OTI cells in mice that were subsequently infected by X31-OVA. Mice that did not receive OTI cells were used as control. Injecting 10^5 WT OTI did not improve response to X31-OVA, as read-out by weight loss. Injecting IFN γ RKO OTI, however, partially reduced weight loss, confirming that IFN γ RKO OTI cells conferred enhanced protection (Fig.S5M).

Reviewers' Comments:

Reviewer #1:

Remarks to the Author:

The authors have successfully revised the manuscript in response to reviewers' comments and suggestions.

Reviewer #2:

Remarks to the Author:

This manuscript is wildly improved and makes a very compelling case. I commend the authors on comprehensively addressing all of my issues experimentally and congratulate them on an outstanding paper

Reviewer #3:

Remarks to the Author:

The authors have responded to all previous concerns, providing new data and analyses that support their central conclusions.

Response to reviewers' comments

We would like to thank the reviewers for their positive evaluation of our manuscript.

Reviewer #1 (Remarks to the Author):

The authors have successfully revised the manuscript in response to reviewers' comments and suggestions.

Reviewer #2 (Remarks to the Author):

This manuscript is wildly improved and makes a very compelling case. I commend the authors on comprehensively addressing all of my issues experimentally and congratulate them on an outstanding paper

Reviewer #3 (Remarks to the Author):

The authors have responded to all previous concerns, providing new data and analyses that support their central conclusions.